# Data-driven modeling reconciles kinetics of ERK phosphorylation, localization, and activity states

Shoeb Ahmed[1], Kyle G Grant[1], Laura E Edwards[2], Anisur Rahman[1], Murat Cirit[1], Michael B Goshe[2] & Jason M Haugh[1,*]

## Abstract

The extracellular signal-regulated kinase (ERK) signaling pathway controls cell proliferation and differentiation in metazoans. Two hallmarks of its dynamics are adaptation of ERK phosphorylation, which has been linked to negative feedback, and nucleocytoplasmic shuttling, which allows active ERK to phosphorylate protein substrates in the nucleus and cytosol. To integrate these complex features, we acquired quantitative biochemical and live-cell microscopy data to reconcile phosphorylation, localization, and activity states of ERK. While maximal growth factor stimulation elicits transient ERK phosphorylation and nuclear translocation responses, ERK activities available to phosphorylate substrates in the cytosol and nuclei show relatively little or no adaptation. Free ERK activity in the nucleus temporally lags the peak in nuclear translocation, indicating a slow process. Additional experiments, guided by kinetic modeling, show that this process is consistent with ERK's modification of and release from nuclear substrate anchors. Thus, adaptation of whole-cell ERK phosphorylation is a by-product of transient protection from phosphatases. Consistent with this interpretation, predictions concerning the dose-dependence of the pathway response and its interruption by inhibition of MEK were experimentally confirmed.

**Keywords** growth factor receptors; mathematical model; mitogen-activated protein kinases; negative feedback; nucleocytoplasmic shuttling

**Subject Categories** Quantitative Biology & Dynamical Systems; Signal Transduction

**Mol Syst Biol. (2014) 10: 718**

## Introduction

Signal transduction networks mediate diverse cellular processes by modulating the cell's gene-regulatory and cytoskeletal systems. In the signaling networks accessed by growth factor and cytokine receptors, the extracellular signal-regulated kinase (ERK) pathway is a principal mode of controlling cell proliferation and other responses, and its aberrant activation contributes to uncontrolled proliferation in the majority of human cancers (Dhillon *et al*, 2007; Roberts & Der, 2007). ERK1 and ERK2 are among the mammalian mitogen-activated protein kinases (MAPKs), which are canonically activated in a three-tiered protein kinase cascade. Upstream of the cascade, cell surface receptors orchestrate signaling through small GTPases, which mediate phosphorylation and activation of Raf-family kinases. In the cascade, enzymatically active Raf proteins phosphorylate and activate MEK1 and MEK2, dual specificity kinases that in turn phosphorylate ERK1 and ERK2 on threonine and tyrosine residues of their homologous TEY motif; only diphosphorylated ERK has elevated kinase activity (Anderson *et al*, 1990). The linear, sequential simplicity of the Raf → MEK → ERK cascade as a pathway motif belies a rich complexity in the regulation of ERK signaling, which has largely emerged from quantitative studies combining experimental measurements and kinetic modeling (Bhalla *et al*, 2002; Schoeberl *et al*, 2002; Sasagawa *et al*, 2005; Fujioka *et al*, 2006; Birtwistle *et al*, 2007; Chen *et al*, 2009; Schilling *et al*, 2009; Wang *et al*, 2009; Cirit *et al*, 2010; Cirit & Haugh, 2012). One important mode of regulation is adaptation of the pathway by ERK-dependent negative feedback, which desensitizes the activity of Raf and/or other upstream components (McKay & Morrison, 2007). Thus, growth factor-stimulated activation of the ERK pathway is typically transient. The kinetics of ERK activation and adaptation have been quantitatively characterized (Cirit *et al*, 2010; Sturm *et al*, 2010; Fritsche-Guenther *et al*, 2011) and have proven to be important for proliferation and cell-fate decisions (Marshall, 1995; von Kriegsheim *et al*, 2009; Chung *et al*, 2010; Albeck *et al*, 2013).

Another complex facet of ERK regulation is its subcellular compartmentalization. Active ERK phosphorylates more than 150 protein substrates, with roughly equal numbers in the cytosol and nucleus (Yoon & Seger, 2006). Thus, nucleocytoplasmic shuttling is an important determinant of ERK function. In quiescent cells, ERK is predominantly localized in the cytoplasm; once fully phosphorylated by MEK, ERK is released from cytoplasmic scaffold proteins and rapidly translocates to the nucleus (Horgan & Stork, 2003;

1  Department of Chemical and Biomolecular Engineering, North Carolina State University, Raleigh, NC, USA
2  Department of Molecular and Structural Biochemistry, North Carolina State University, Raleigh, NC, USA
   *Corresponding author. Tel: +1 919 513 3851; Fax: +1 919 515 3465; E-mail: jason_haugh@ncsu.edu

Burack & Shaw, 2005; Costa *et al*, 2006; Lidke *et al*, 2010; Marchi *et al*, 2010; Zehorai *et al*, 2010). Both passive and energy-dependent mechanisms of ERK nuclear translocation have been proposed, with the latter mediated by the shuttling protein Importin-7 and nuclear Ran (Plotnikov *et al*, 2011). While in the nucleus, active ERK modifies substrates and is dephosphorylated and/or exported to the cytoplasm to complete the cycle; evidence suggests that nuclear ERK is exported rapidly, irrespective of its phosphorylation state (Horgan & Stork, 2003). In live-cell fluorescence microscopy experiments, nuclear localization of ERK typically exhibits a transient peak, sustained oscillations, or damped oscillations with time (Costa *et al*, 2006; Sato *et al*, 2007; Cohen-Saidon *et al*, 2009; Shankaran *et al*, 2009); these kinetics are cell- and stimulus-specific (Shankaran *et al*, 2009; Weber *et al*, 2010) and have been linked to the aforementioned negative feedback regulation of the pathway. Yet, with a couple of notable exceptions (Fujioka *et al*, 2006; Shankaran *et al*, 2009), kinetic models describing both adaptation and compartmentalization of ERK signaling have not been considered. The implicit assumption, which if valid would obviate the need for such integration, is that nuclear localization of ERK mirrors the kinetics of ERK activation in the cytosol. Taken a step farther, the often-unstated assumption is that whole-cell phosphorylation and nuclear translocation of ERK mirror the kinase activity of ERK in both compartments.

In this study, we combine single-cell ERK localization and activity measurements to demonstrate that this assumption is not justified in general. In fibroblasts stimulated with a high dose of platelet-derived growth factor (PDGF), nuclear translocation of ERK exhibits the typical transient kinetics, whereas neither active ERK in the cytosol nor in the nucleus show dramatic adaptation during the same period of observation. Strikingly, the accumulation of free, active ERK in the nucleus temporally lags its overall nuclear translocation. These data, together with whole-cell biochemical measurements, are reconciled by a mathematical model accounting for interactions between ERK and its substrates in the cytosol and nucleus. The interpretation is that phosphorylation of those substrates by ERK, and not feedback adaptation, accounts for the dramatic overshoot of ERK phosphorylation and nuclear localization. The model yields qualitative predictions regarding the pathway kinetics under various stimulation and inhibition conditions, which we tested and confirmed experimentally. There are ample indications in the literature that interactions with substrates can control localization of ERK (Caunt & McArdle, 2010; Lidke *et al*, 2010) and its available kinase activity (Tanoue *et al*, 2000; Bardwell *et al*, 2003; Kim *et al*, 2011a,b). This work implicates how those interactions manifest in the temporal dynamics of ERK signaling in mammalian cells, and it sheds new light on the role of negative feedback in shaping input-output relationships when nucleocytoplasmic shuttling is involved.

## Results

### Growth factor-stimulated phosphorylation and nuclear translocation of ERK show dramatic adaptation, whereas the kinase activities of ERK in the cytosol and nucleus do not

As outlined in the Introduction section, a host of published studies have reported on the kinetics of growth factor-stimulated ERK phosphorylation and nucleocytoplasmic shuttling. The consensus finding is that these hallmarks of ERK activation exhibit adaptation within the first hour of stimulation, with a peak magnitude that is much greater than the quasi-steady, plateau level achieved later on. Accordingly, in mouse fibroblasts stimulated with 1 nM PDGF, quantitative immunoblotting using phospho-specific antibodies revealed MEK and ERK phosphorylation time courses with high degrees of adaptation (Fig 1A and B and Supplementary Fig S1). Even with the limited temporal resolution of these measurements, phosphorylation of MEK clearly peaks earlier than that of ERK. ERK phosphorylation decays more slowly and adapts almost completely within 1–2 h (Fig 1B). These findings were corroborated by label-free quantitative mass spectrometry, showing near-complete adaptation in the amount of diphosphorylated ERK2 pTEpY peptide (Fig 1C, top). By comparison, the amount of mono-phosphorylated ERK2 pTEY peptide exhibited more subtle adaptation (Fig 1C, bottom), and the mono-phosphorylated ERK2 TEpY peptide showed even less (Supplementary Fig S2, which also shows similar phosphorylation kinetics for ERK1 and ERK2). Consistent with the notion that the active form of ERK adapts, individual cells expressing a GFP-ERK1 fusion construct (Shankaran *et al*, 2009) showed transient nuclear translocation responses to PDGF stimulation, reaching a peak value within 10 min followed by a dramatic decay (Fig 1D). At least on a qualitative basis, such data do little to challenge the aforementioned assumption that nuclear translocation of ERK tracks its phosphorylation and enzymatic activity, an interpretation supported by photobleaching measurements demonstrating rapid nucleocytoplasmic exchange of ERK (Burack & Shaw, 2005).

To more critically test the relationship between nuclear localization and kinase activity of ERK, we employed a live-cell imaging strategy with cells co-expressing a Förster resonance energy transfer (FRET)-based ERK kinase activity reporter (EKAR) construct (Harvey *et al*, 2008) and an mCherry-ERK2 fusion (Fig 2). The EKAR biosensor contains a phospho-binding (WW) domain, a substrate sequence corresponding to Cdc25C Thr48, and a docking site for ERK FXF (DEF) motif, flanked by the FRET pair Cerulean and Venus. The specificity of this biosensor for the ERK pathway in our cells was confirmed using inhibitors of MEK (U0126) (Duncia *et al*, 1998) and ERK (Hancock *et al*, 2005), which completely blocked the response to PDGF (Supplementary Fig S3). To assess localized ERK activity we used both cytosolic and nuclear EKAR probes (Harvey *et al*, 2008), each in tandem with mCherry-ERK2 in the same cells (Fig 2A–C and D–F, respectively; Supplementary Fig S4 shows quantification for the individual cells). In both cases, the kinetics of PDGF-stimulated mCherry-ERK2 nuclear localization showed a transient, adaptive response, consistent with the kinetics seen in cells expressing GFP-ERK1 (Fig 2B and E; compare to Fig 1D).

In stark contrast with ERK nuclear translocation kinetics, neither the ERK activity in the cytosol nor in the nucleus showed more than modest adaptation as reported by EKAR FRET (Fig 2C and F). The average cytosolic EKAR response peaked rapidly (4–5 min after stimulation) and decayed by only about one-third of its peak value above basal thereafter (Fig 2C). Even more distinct was the nuclear EKAR response, which showed relatively slow accumulation over a period of approximately 20 min and little or no adaptation (Fig 2F); qualitatively similar kinetics for nuclear EKAR, in relation to the nuclear translocation of ERK2, were elicited by stimulation with

                                                                 

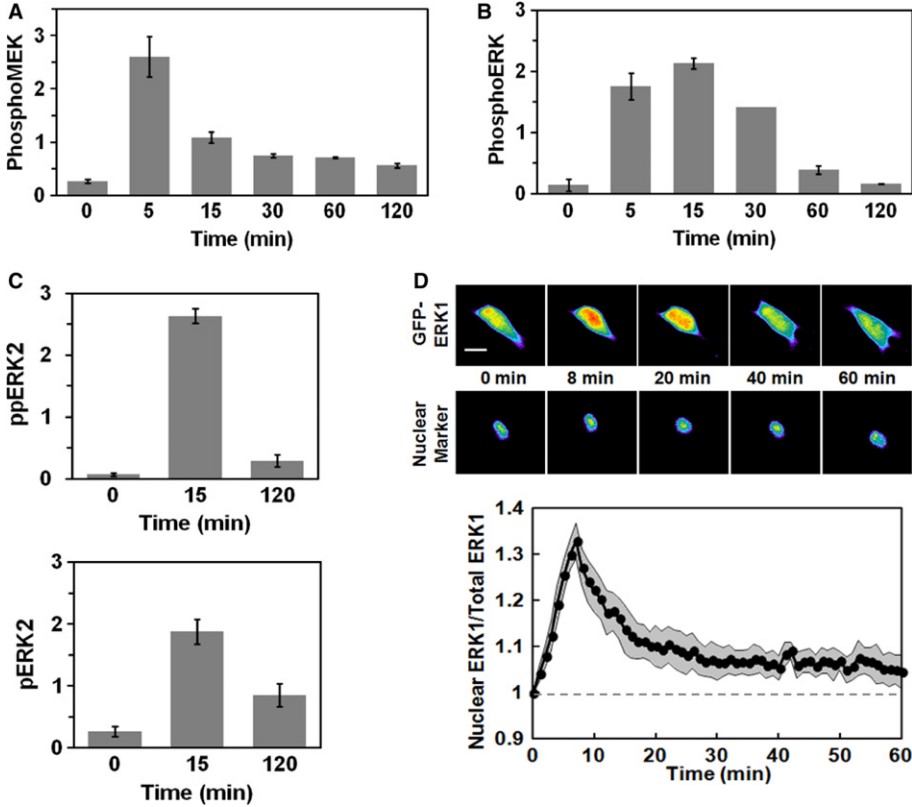

**Figure 1.    Transient phosphorylation and nuclear localization of ERK.**

A, B    NIH 3T3 cells were stimulated with 1 nM PDGF, and phosphorylation of MEK1/2 [(A); adapted from (Cirit *et al*, 2010)] and ERK1/2 (B) were assessed by quantitative immunoblotting along with total ERK1/2 as a loading control. Values are reported as mean ± s.e.m. in arbitrary units ($n \geq 3$).

C    Label-free quantitative mass spectrometry results show relative amounts of diphosphorylated ERK2 (pTEpY, top) and mono-phosphorylated ERK2 (pTEY, bottom) in NIH 3T3 cells stimulated maximally with PDGF for the indicated times. Values are reported as mean ± s.e.m. ($n = 3$).

D    NIH 3T3 cells expressing GFP-ERK1 were monitored by epifluorescence microscopy during maximal PDGF stimulation. The pseudocolor montage shows redistribution of fluorescence in a representative cell (scale bar, 20 μm). Nuclei were labeled using a genetically encoded nuclear marker. Mean nuclear localization of ERK1, normalized by its initial value, is plotted as a function of time (solid line; $n = 10$); the gray region reports the 95% confidence interval.

another growth factor, FGF-2 (Supplementary Fig S5). To validate this approach we used immunofluorescence staining to assess PDGF-stimulated phosphorylation of an endogenous substrate of ERK in the nucleus, Elk-1. Consistent with the nuclear EKAR response, mean Elk-1 phosphorylation in nuclei peaked after roughly 20 min; thereafter, there was only a subtle dip in the phosphorylation level (Fig 2G). Phosphorylation of a cytosolic ERK substrate, MEK1 Thr292 (Gardner *et al*, 1994; Xu *et al*, 1999), was also evaluated. As expected, this readout showed a rapid response and little adaptation within the first 60 min of PDGF stimulation (Supplementary Fig S6).

**The rate-controlling process affecting ERK dynamics is linked to the availability of free, active ERK in the nucleus**

The apparently disparate kinetics of ERK catalytic activity as compared with ERK phosphorylation and nuclear translocation suggest the influence of a relatively slow process governing the accumulation of nuclear EKAR FRET, which occurs in concert with adaptation of ERK nuclear localization. One plausible explanation is that nuclear EKAR is dephosphorylated slowly, in which case its

response would not track an earlier peak in nuclear ERK activity, if one were present. In theory, if the affinity of the intramolecular interaction enabling FRET in the phosphorylated EKAR chain were too high, such a sluggish biosensor response would be expected (Haugh, 2012).

To exclude this possibility, we stimulated cells with PDGF until the response was quasi-steady, followed by addition of the MEK inhibitor, U0126. Both cytosolic and nuclear EKAR signals decayed completely and rapidly after addition of U0126, with $t_{1/2}$ values approximately 2 min (the temporal resolution of these experiments) or less (Fig 3A and B). These results are consistent with previous observations that ERK activity and nuclear localization require continuous activation of the MEK-ERK pathway (Burack & Shaw, 2005). More importantly, the rapid decay kinetics rule out the hypothesis outlined above, indicating instead that the availability of active ERK is rate-limiting for PDGF-stimulated accumulation of EKAR signal; in other words, we can safely assume that the EKAR substrates rapidly equilibrate with free, diphosphorylated ERK. In other experiments we replaced U0126 with okadaic acid, an inhibitor of multiple phosphatases. Both the cytosolic and nuclear EKAR FRET readouts

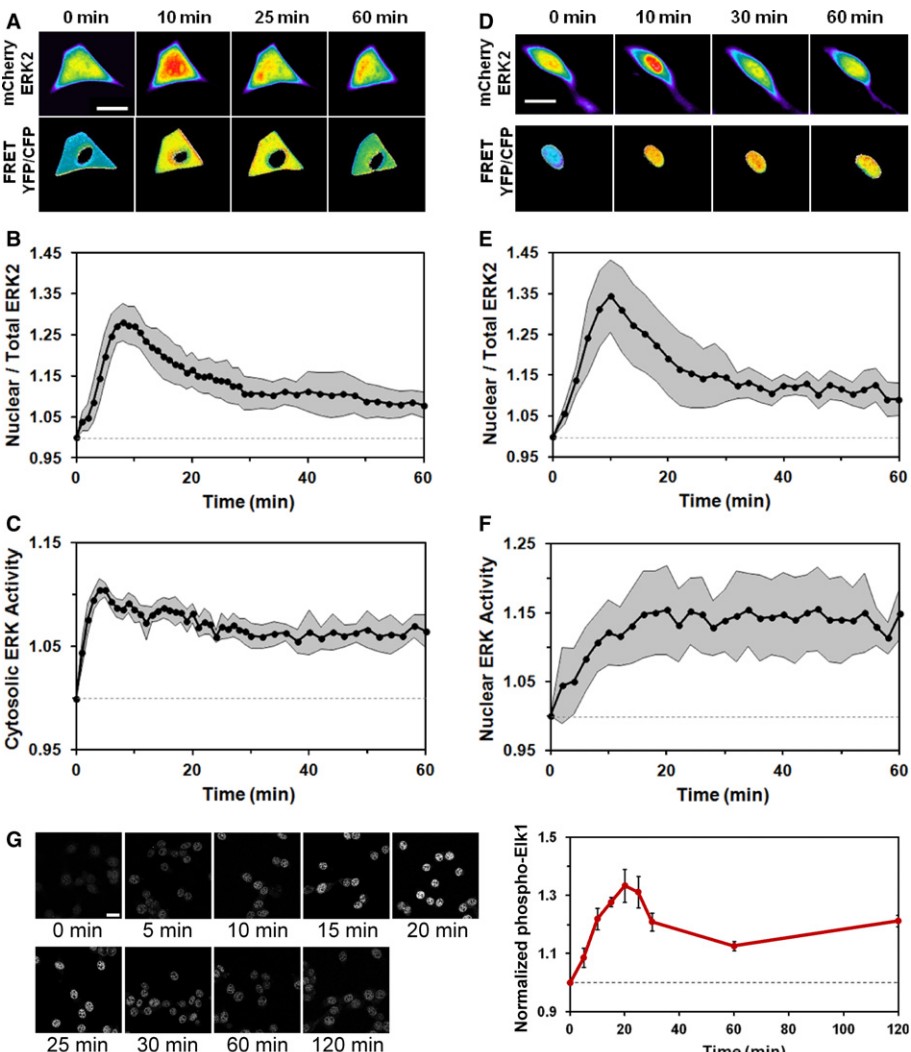

**Figure 2.  Growth factor-stimulated ERK kinase activities in the cytosol and nucleus are kinetically ordered and do not show dramatic adaptation.**

A–C  NIH 3T3 cells co-expressing mCherry-ERK2 and cytosolic ERK kinase activity reporter (EKAR) were observed by epifluorescence microscopy during maximal PDGF stimulation. The pseudocolor montage (A) shows representative image data for this experiment (scale bar = 20 μm). Mean time courses of ERK2 nuclear localization (B) and cytosolic ERK activity (C) were measured in tandem (mean ± 95% confidence interval, *n* = 8).

D–F  NIH 3T3 cells co-expressing mCherry-ERK2 and nuclear EKAR were observed by epifluorescence microscopy during maximal PDGF stimulation. The pseudocolor montage (D) shows representative image data for this experiment (scale bar = 20 μm). Time courses of nuclear localization (E) and nuclear ERK activity (F) were measured in tandem (mean ± 95% confidence interval, *n* = 6).

G  Montage of a representative cell shows immunofluorescence staining of phosphorylated Elk-1, an endogenous substrate of ERK in the nucleus, upon maximal PDGF stimulation (scale bar = 20 m).  The plot shows the time course of phosphorylated, nuclear Elk-1 (note that the time axis is expanded relative to E&F). Values were normalized by the initial value and are reported as mean ± s.e.m. (*n* = 3).

increased further after okadaic acid treatment (Fig 3C and D), confirming that the biosensor signal remains in the dynamic range during PDGF stimulation.

These results indicate that ERK dephosphorylation is not a rate-controlling process in our cells; i.e. it cannot explain why the adaptation of ERK phosphorylation noticeably lags behind that of MEK phosphorylation (Fig 1A and B). Previous analyses that did not account for nucleocytoplasmic shuttling, including our own (Schoeberl *et al*, 2002; Sasagawa *et al*, 2005; Cirit *et al*, 2010), implicitly rely on slow ERK dephosphorylation to explain the lag.

**A kinetic model including ERK interactions with cytosolic and nuclear substrates reconciles observed ERK phosphorylation, localization, and activity responses**

The data presented above indicates that there is a temporal lag between nuclear translocation of ERK and its available activity in the nucleus. Hence we reasoned that competitive interactions between ERK and its many substrate proteins, which have been shown to be important during *Drosophila* embryogenesis (Kim *et al*, 2011a,b), might mechanistically account for the distinct temporal

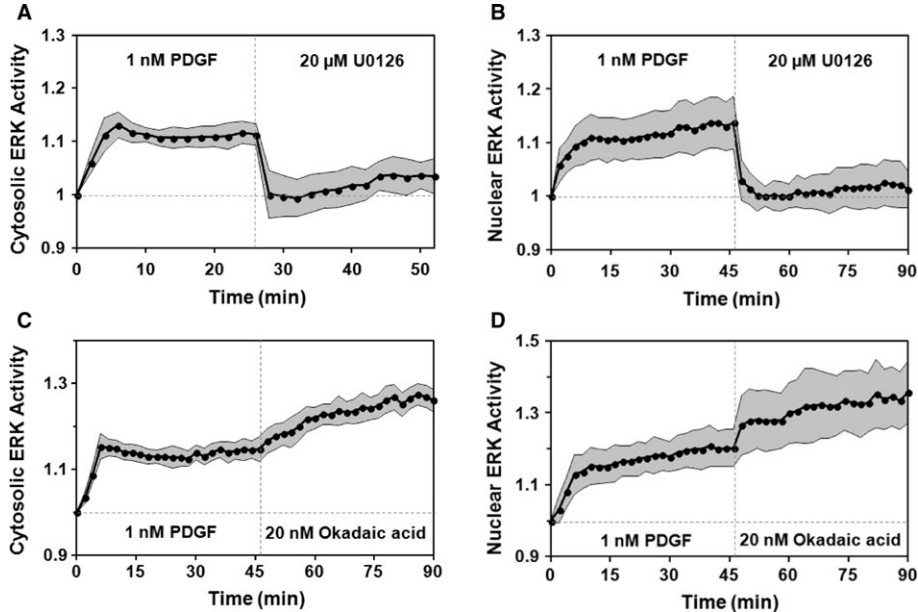

**Figure 3. ERK activity measurements reliably track the availability of free, active ERK.**

NIH 3T3 cells expressing FRET-based EKAR biosensors were monitored by epifluorescence microscopy during maximal PDGF stimulation.

A, B Mean time courses of cytosolic [(A), $n = 10$] and nuclear [(B), $n = 9$] ERK activities show responses to maximal PDGF treatment followed by addition of the MEK inhibitor, U0126.

C, D Mean time courses of cytosolic [(C), $n = 9$] and nuclear [(D), $n = 9$] ERK activity in response to maximal PDGF stimulation followed by addition of the phosphatase inhibitor, okadaic acid. In all of the plots, the gray regions report 95% confidence intervals.

features of our data set. To evaluate this hypothesis, we formulated minimal kinetic models of ERK dynamics in the cytosol and nucleus (Supplementary Text S1). The models were judged based on their abilities to globally fit the data set using a Monte Carlo, parameter set ensemble approach (Brown & Sethna, 2003; Wang *et al*, 2009; Cirit *et al*, 2010; Hao *et al*, 2012).

In accord with our hypothesis, a model accounting for MEK and ERK phosphorylation in the cytosol, nucleocytoplasmic shuttling of ERK, and ERK-substrate interactions in both compartments (Fig 4A) successfully reconciled the biochemical (Fig 4B–E) and live-cell imaging (Fig 4F–H) measurements and the fast decay kinetics seen in cells treated with PDGF followed by MEK inhibition (Fig 4I and J). A key aspect of the fit is that the total amount of diphosphorylated ERK (as measured in a cell lysate) includes both the free and substrate-bound pools. The fit shows that the kinetics of free, diphosphorylated ERK are actually better reflected in the accumulation of mono-phosphorylated ERK, for which the slow step is the liberation of substrate-bound ERK followed by rapid dephosphorylation and, in the case of nuclear ERK, export to the cytosol.

A "control" model neglecting the influence of substrate interactions was tested, and it fails on a qualitative level to fit the data set (Supplementary Fig S7). With negligible substrate interactions, the whole-cell level of diphosphorylated ERK is equal to the sum of the active ERK in the cytosol and nucleus, as read out by the EKAR measurements. The (arbitrarily weighted) sum of two time courses, each of which showing a low degree of adaptation, will exhibit the same property; therefore, the comparison of the EKAR kinetics with those of ERK diphosphorylation imposes an irreconcilable conflict for the "control" model, which is systematically constrained to generate

ERK phosphorylation, nuclear translocation, and nuclear activity time courses that have approximately the same shape (Supplementary Fig S7).

An alternative "buffering" model was also evaluated. Rather than assuming that substrate interactions control the availability of active ERK in the nucleus, this model neglected substrate interactions and considered instead that diphosphorylated ERK entering the nucleus must be liberated from its interaction with the shuttling protein, Importin-7 (Stewart, 2007; Plotnikov *et al*, 2011). This model variation is implemented with the hypothesis that release of Importin7-bound ERK might be rate-controlling for accumulation of free, active ERK in the nucleus; however, like the "control" model, it fails to fit the data (Supplementary Fig S8). This model fails because the influence of the assumed release mechanism applies to ERK entering the nucleus at any time, and thus the amount of nuclear ERK in the importin-bound state cannot be highly transient unless the amount of free, active ERK in the cytosol (rapidly equilibrated with cytosolic EKAR) is also. For the same reason, adding the release mechanism to the successful model including substrate interactions did not demonstrably improve the quality of fit (Supplementary Fig S9).

## Substrate interactions as a transient buffer of diphosphorylated ERK

The intuitive interpretation offered by the successfully fit model is as follows. ERK is phosphorylated and achieves a quasi-steady state in the cytosol fairly rapidly, as seen in the cytosolic EKAR response, while ERK steadily accumulates in the nucleus. There, active ERK is initially buffered by interactions with nuclear substrates. The notion

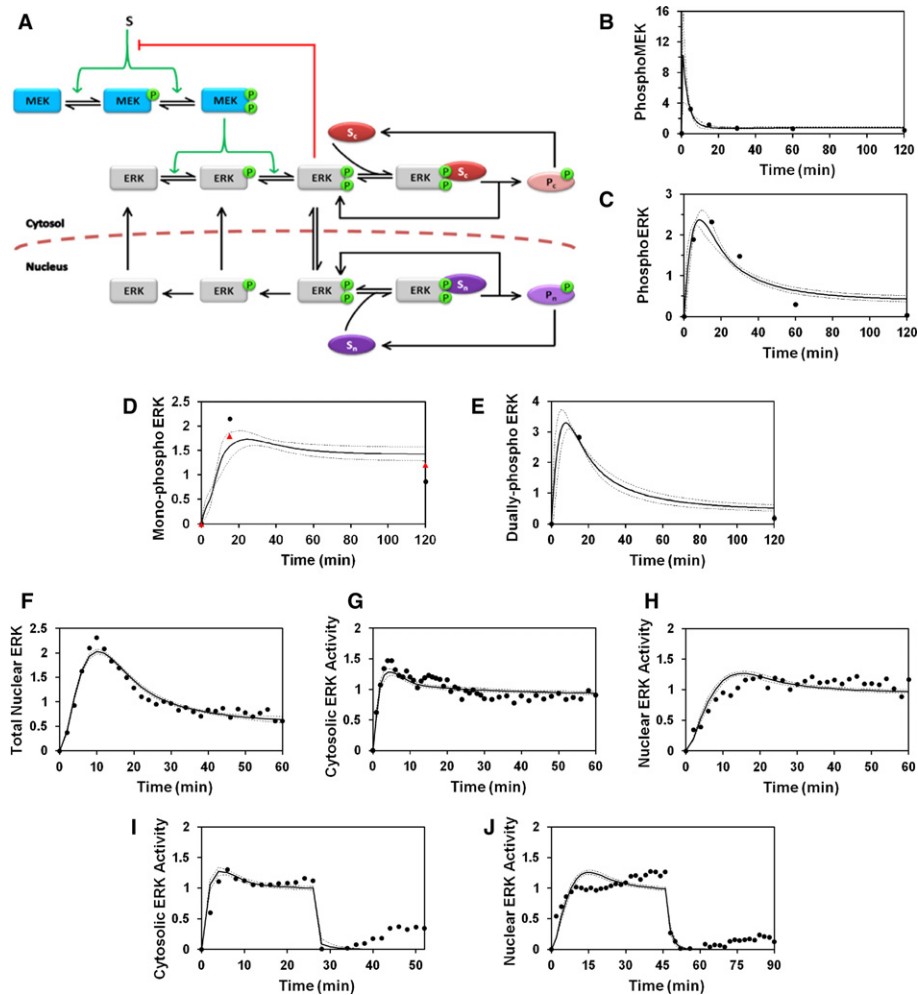

**Figure 4.   A kinetic model reconciles observed ERK phosphorylation, localization, and activity responses.**

A    Schematic of a simple kinetic model including cytosolic and nuclear substrates that bind to and are phosphorylated by active ERK.

B–J   Means of calculated time courses (solid lines), representing a global ensemble fit to the data, are plotted along with the means of the corresponding experimental measurements (black circles). Broken lines report mean ± s.d. of the model output for all parameter sets in the ensemble (*n* = 10 000). The data fit for maximal PDGF stimulation conditions comprises MEK (B) and ERK (C) phosphorylation measured by immunoblotting, mono-phosphorylated ERK2 [pTEY, (D)] and diphosphorylated ERK2 (E) measured by mass spectrometry, nuclear localization of ERK (F), and cytosolic (G) and nuclear (H) ERK activities. Also fit were the experiments in which cytosolic (I) and nuclear (J) ERK activities were monitored in cells maximally stimulated with PDGF followed by MEK inhibition.

Data information: In (D), the data for the other mono-phosphorylated form of ERK2, TEpY, are also shown (red triangles).

Source data are available online for this figure.

that those interactions are competitive means that most of the nuclear ERK is bound and also protected from dephosphorylation during this phase (Bardwell *et al*, 2003; Kim *et al*, 2011a,b). As time goes on, ERK phosphorylates those substrates, freeing up a higher fraction of the active ERK in the nucleus while also permitting its dephosphorylation and export. Thus, the adaptation of total ERK in the nucleus reflects a reduction in the fraction of diphosphorylated ERK that is substrate-bound (and anchored in the nucleus) versus freely available.

This conceptual model implies that there is a certain high-affinity substrate, or a subset of such substrates, that buffers the free pool of active ERK and thus affects the phosphorylation of all other ERK substrates (Rowland *et al*, 2012). For the buffering to be transient,

the dominant substrate(s) must be phosphorylated gradually and, ultimately, with high stoichiometry (an alternate explanation, not explored here, is that the phosphorylated substrate is degraded). At the other extreme, one might expect to identify substrates with much lower affinity for ERK and relatively fast dephosphorylation kinetics (as assumed for the EKAR probes), which equilibrate rapidly with the free pool of active ERK. Consistent with this view, we find that the shape of the phosphorylation time course of Elk-1, as quantified in Fig 2G, is intermediate between the predicted high- and low-affinity extremes for the nuclear compartment (Supplementary Fig S10).

A deeper analysis of the ensemble of fit parameter sets provides additional insight. Reasoning that substrate interactions are the

key aspect of the fit, we assessed the maximum buffering strength of each substrate as a parameter grouping defined as the ratio of total substrate concentration to the associated Michaelis constant, $K_m$ (Supplementary Text S1). A scatter plot of nuclear vs. cytosolic buffering strengths for each parameter set in the ensemble revealed two distinct modes by which a good fit was achieved (Fig 5). In one mode, a particular buffering ratio for nuclear substrate (~20) is sufficient to explain the data, while cytosolic substrate offers minimal buffering (or, at least, minimal cytosolic anchoring). In the other mode, cytosolic substrate has a particular buffering ratio (also ~20), while in the nucleus the buffering ratio still must be significant, but it need only exceed unity. Analysis of the model output for five selected parameter sets (indicated by the arrows in Fig 5) shows that nuclear buffering is required to explain the kinetic delay between overall nuclear translocation and availability of nuclear ERK activity, while buffering in either compartment is sufficient to explain the apparent adaptation of diphosphorylated ERK relative to availability of ERK activity on a whole-cell basis (Fig 5).

A key insight that emerges from the data and analysis described above is that the transience of ERK phosphorylation does not simply reflect a slow dephosphorylation of ERK following the rapid adaptation of MEK by negative feedback. This implies that adaptation of MEK is not necessary, and that substrate interactions are sufficient, to explain the adaptation of ERK phosphorylation. To show this, we fit to the ERK data a variant of the model in which MEK activity is held constant. Although the fit was, in certain respects, not as good as the fit shown in Fig 4 (e.g. in fitting the cytosolic EKAR data), this variation successfully captured the apparent adaptation of the ERK phosphorylation and nuclear translocation responses (Supplementary Fig S11).

## Experimental confirmation of qualitative model predictions

Having established a model-guided, mechanistic interpretation of the available data, we sought to test associated predictions. Three related predictions concerned the kinetics and steady-state response of the system as a function of the input signal strength (Fig 6). First, we compared PDGF doses that yield maximal (1 nM) and roughly half-maximal (30 pM) phosphorylation of ERK (Supplementary Fig S1).

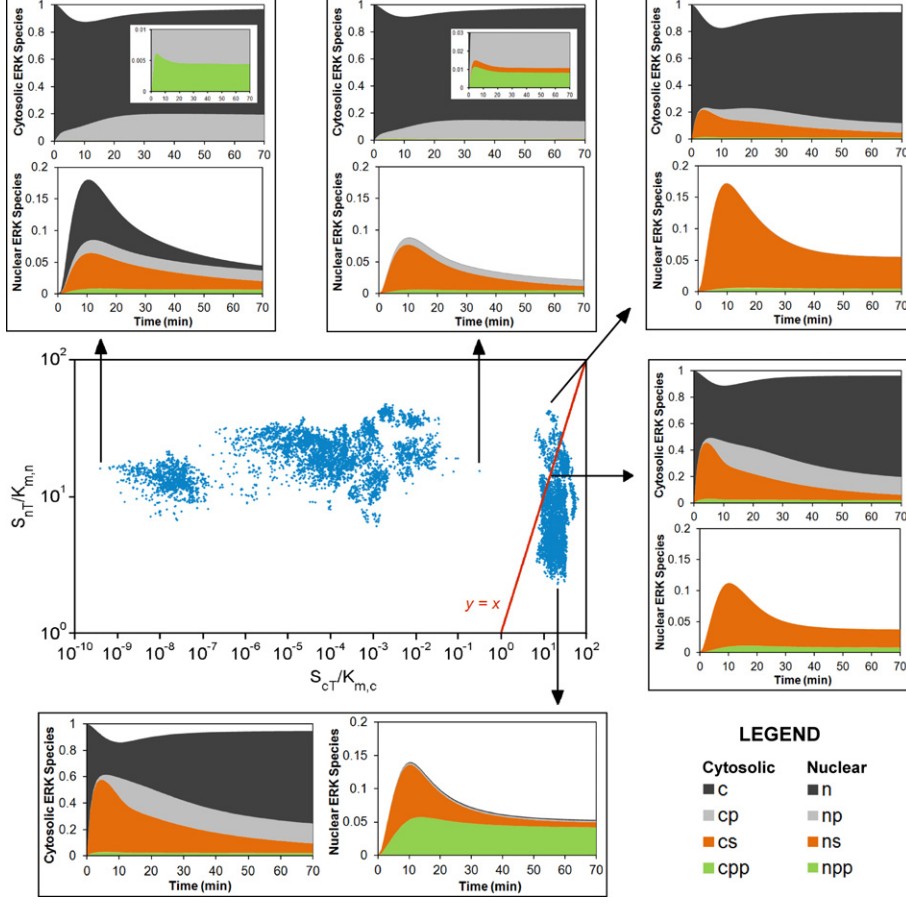

**Figure 5.  The strengths of ERK-substrate interactions in the cytosol and nucleus largely explain the model fit to the data.**
Substrate buffering strength (total substrate concentration divided by the corresponding $K_m$) in the cytosol and in the nucleus are plotted for each of the 10,000 parameter sets in the ensemble (center). Breakdowns of ERK species in the cytosol and nucleus are shown for 5 of the parameter sets as connected by the arrows, selected based on their relative substrate buffering strengths. c/n, cytosolic/nuclear unphosphorylated ERK; cp/np, cytosolic/nuclear mono-phosphorylated ERK; cpp/npp, cytosolic/nuclear diphosphorylated ERK; cs/ns, cytosolic/nuclear substrate-bound ERK.

Both doses elicit transient, peaked ERK phosphorylation kinetics (Fig 6A). Using a single parameter set picked from the ensemble and comparing half-maximal ($S = 0.5$) versus maximal ($S = 1$) input strengths, the model output shows qualitative agreement with these data (Fig 6B); other parameter sets yielded similar results with modest adjustment of the submaximal $S$ value (e.g. in the range of $S = 0.3–0.6$, because the saturable relationship between PDGF dose and $S$ is approximate). These results suggest that the transient buffering of phosphorylated ERK, on a whole-cell basis, is consistent between the two stimulation conditions. Second, we measured nuclear translocation of ERK2 stimulated by the lower PDGF dose; after 30 min, the PDGF concentration was adjusted to 1 nM. In both the experiments and model output, ERK2 nuclear translocation shows a muted peak for the low dose and little or no apparent

increase after the increase in PDGF concentration (Fig 6C and D; Supplementary Fig S12A shows quantification for the individual cells, and Supplementary Fig S12B presents parallel controls in which the 30 pM stimulation period was replaced by mock stimulation with buffer). The lack of dose responsiveness here cannot be attributed to adaptation of the upstream pathway; guided by a third model prediction, we confirmed that free, active ERK in the cytosol — the driving force for nuclear translocation — almost doubles after the PDGF dose is increased (Fig 6E and F).

In the context of the model, these results are reconciled as follows. Increasing the input strength yields greater ERK phosphorylation by active MEK in the cytosol, resulting in greater influx of ERK into the nucleus. This increased influx is approximately offset by an increase in efflux, as the dynamic equilibrium is shifted

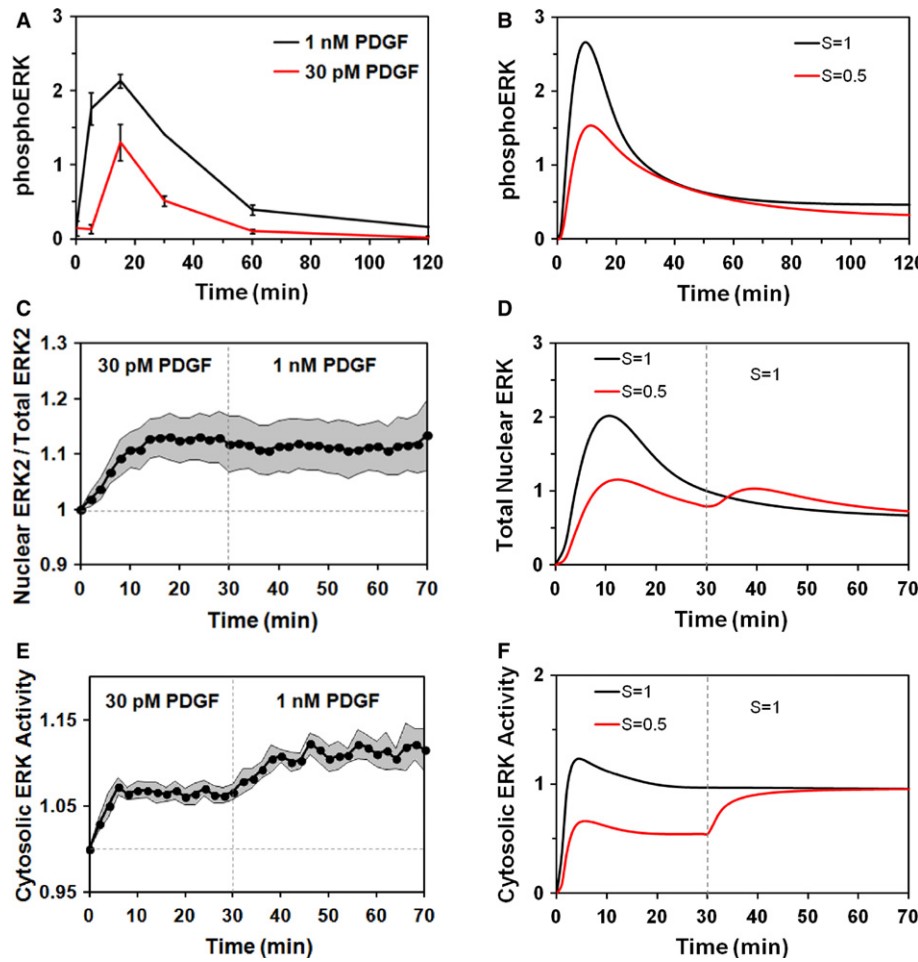

**Figure 6.  Testing model predictions for half-maximal stimulation.**
A representative parameter set from the ensemble was used to predict ERK phosphorylation, nuclear translocation, and activity kinetics at half-maximal input strength ($S = 0.5$) relative to maximal PDGF stimulation ($S = 1$). These were compared to experimental measurements using 30 pM PDGF as the submaximal dose.

A    ERK phosphorylation was assessed by quantitative immunoblotting (mean ± s.e.m., $n = 3$).
B    Predicted ERK phosphorylation.
C    Nuclear localization of mCherry ERK2 was stimulated by 30 pM PDGF followed by 1 nM PDGF (mean ± 95% confidence interval, $n = 8$).
D    Predicted nuclear translocation of ERK.
E    Cytosolic ERK activity was measured for the same stimulation protocol as in (C) (mean ± 95% confidence interval, $n = 11$).
F    Prediction of free, active ERK in the cytosol.

towards greater phosphorylation of ERK substrate(s) and thus less substrate to anchor ERK in the nucleus.

A fourth and wholly different prediction tests the concept that diphosphorylated ERK is transiently protected from phosphatase activity. We reasoned that interrupting the pathway (by inhibiting MEK) during the early, transient period of growth factor-stimulated ERK dynamics would result in a slower decay of the system as compared with the rapid decay documented when the pathway is blocked later (Fig 3). To test this, we assessed the decay of both ERK2 nuclear translocation and free nuclear activity of ERK (Fig 7A and B, respectively) after 10 min of maximal PDGF stimulation followed by MEK inhibition. For comparison, we applied in paired experiments the previous stimulation/inhibition protocol, with addition of the MEK inhibitor at pseudo-steady state. Consistent with model calculations, the measured decays for individual cells following the earlier addition of MEK inhibitor were noticeably slower; following MEK inhibition imposed 10 min post-stimulation, the nuclear dynamics decayed with $t_{1/2}$ values (6.5 ± 3.2 and

4.8 ± 0.5 min for nuclear localization and activity, respectively) that are approximately 3-fold longer than when inhibition was imposed later (1.7 ± 0.5 and 1.4 ± 0.4 min, respectively) (Fig 7). Both comparisons are significant, with $P < 10^{-3}$ by two-tailed $t$-test.

## Discussion

Our analysis of growth factor-stimulated ERK dynamics encompasses negative feedback, nucleocytoplasmic shuttling, and substrate interactions, all of which are necessary to fully reconcile diverse biochemical and live-cell microscopy data addressing phosphorylation, localization, and activity states of ERK. Surprisingly, we found that negative feedback is not necessary to explain the apparent transience of ERK phosphorylation and nuclear localization responses. Rather, considering the known competitive interactions of diphosphorylated ERK with substrates and phosphatases (Bardwell *et al*, 2003; Kim *et al*, 2011b), the interpretation is that

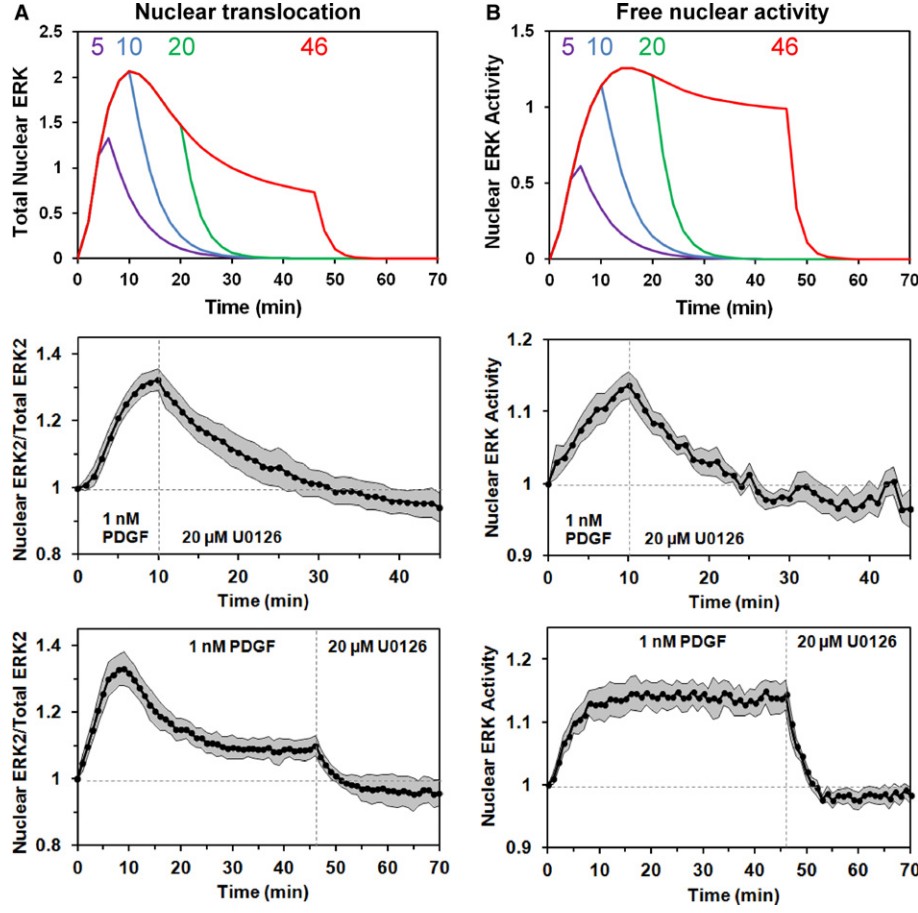

**Figure 7.  Transient buffering is reflected in decay of the system following MEK inhibition.**

A, B  Nuclear translocation of ERK (A) and free, active ERK in the nucleus (B) were predicted (mean of ensemble predictions) for maximal PDGF stimulation followed by MEK inhibition imposed at different time points (5, 10, 20, and 46 min after PDGF stimulation, as indicated) (top panels). NIH 3T3 cells co-expressing mCherry-ERK2 and nuclear EKAR biosensor were treated with 1 nM PDGF followed by U0126 MEK inhibitor to test the predictions. The data are presented as mean ± 95% confidence interval. Middle panels: kinetics of nuclear localization [(A), $n = 10$] and nuclear ERK activity [(B), $n = 7$; of the 10 cells in (A), these had suitable EKAR expression] for MEK inhibition imposed 10 min after PDGF stimulation. Bottom panels: kinetics of nuclear localization [(A), $n = 9$] and nuclear ERK activity [(B), $n = 9$] for MEK inhibition imposed 46 min after PDGF stimulation.

transient ERK dynamics are a consequence of relatively rapid ERK activation and nuclear import followed by slower equilibration of ERK substrate phosphorylation/dephosphorylation (or possibly degradation of the substrate). With the availability of active ERK transiently buffered by substrate binding, the pools of free, active ERK in the cytosol and nucleus show little adaptation. This work implicates mechanisms that affect the temporal and compartmental dynamics of ERK signaling in mammalian cells.

Our measurements and model-guided analysis illustrate that the kinetic relationships among ERK phosphorylation, localization, and activity vary with the magnitude of the input. A lower dose of growth factor still elicits a transient phosphorylation response in our cells, but the apparent peak of nuclear localization is lost. When the concentration of growth factor was subsequently increased to a maximal dose, there was little or no increase in the level of nuclear ERK. This apparent adaptation is not consistent with desensitization of ERK phosphorylation in the cytosol; rather, the data and model-guided analysis strongly suggest that the "adaptation" is intrinsic to the nuclear dynamics, caused by a redistribution of ERK from the substrate-anchored state to the unbound state. More generally, the analysis shows that measurable ERK dynamics are sensitive to the degree of buffering by ERK substrates (Fig 5). We note that expression of either of the EKAR substrates does not noticeably perturb ERK in our cells, as judged from the consistent nuclear translocation kinetics measured in parallel with EKAR responses or in the absence of EKAR expression. This observation underscores the conclusion that only the substrates with the highest buffering strengths affect ERK dynamics. The kinetics and extent of substrate phosphorylation, which depend on the expression of phosphatases that reverse the action of ERK, are just as important (Supplementary Fig S10). Therefore, qualitative, context-dependent differences in observable kinetics should be expected across cell lineages and culture conditions that affect the expression levels of ERK substrates and phosphatases. Indeed, in EGF-stimulated PC12 cells, free ERK activities in the cytosol and nucleus were found to be transient, with no apparent lag between the cytosolic and nuclear responses (Herbst et al, 2011).

This work has forced us to reevaluate how the ERK pathway responds to growth factor stimulation and the role of negative feedback in that response. In our view the true output of the pathway is not phosphorylated ERK nor even ERK kinase activity (as one would measure in a dilute cell extract or after immuno-capture), but rather the amount of active ERK that is free to inter-act with its many substrates in cells. The data indicate that this output, as quantified for cytosolic and nuclear ERK, is highly damped relative to the transience of the classic biochemical read-outs or, when the PDGF dose is saturating, of total ERK localiza-tion in the nucleus. From a process control perspective, one might conjecture that the pathway, subject as it is to strong negative feedback affecting MEK, is tuned so that mobilization of active ERK is not accompanied by dramatic overshoot. Another charac-teristic shown in the data is a kinetic hierarchy in which ERK phosphorylation of cytosolic substrates precedes phosphorylation of nuclear substrates. While transient buffering explains these temporal aspects, negative feedback is critical for the steady-state properties of the system, characterized by linear sensitivity (Joslin et al, 2010; Sturm et al, 2010) and, in our cells, a dynamic range in the sub-nanomolar regime of PDGF-BB or FGF-2 concentration.

Thus, in certain contexts the complex regulatory structure of the ERK pathway apparently yields simple (linear, damped) outputs. This work offers a framework for understanding the basis of these properties and comparing them across different experimental contexts.

# Materials and Methods

### Cell culture, plasmids, and reagents

NIH 3T3 mouse fibroblasts were acquired from American Type Culture Collection (Manassas, VA) and cultured at 37°C, 5% $CO_2$ in Dulbecco's Modified Eagle Medium supplemented with 10% fetal bovine serum, 2 mM L-glutamine, and the antibiotics penicillin and streptomycin. All tissue culture reagents were purchased from Invitrogen (Carlsbad, CA).

Stable NIH 3T3 lines expressing GFP-ERK1 or mCherry-ERK2 were established by retroviral infection and puromycin selection, as described previously (Weiger et al, 2010). Retroviral plasmids encoding GFP-ERK1 and mRFPnuc nuclear marker were kind gifts from Dr. Steven Wiley (Pacific Northwest National Laboratory) (Shankaran et al, 2009). To make the mCherry-ERK2 plasmid, we obtain murine ERK2 cDNA (IMAGE Clone 6468233) from Thermo Scientific Open Biosystems (Waltham, MA). The ERK2 coding sequence was PCR-amplified with flanking HindIII sites, and this product was ligated into a HindIII-digested pBM-IRES-puro mCherry-AktPH plasmid (Welf et al, 2012). Cytoplasmic and nuclear ERK kinase activity reporter (EKAR, Cerulean-Venus) plasmids (Harvey et al, 2008) were obtained from Addgene (Cambridge, MA) and transfected into cells by lipofection.

Antibodies recognizing phospho-MEK1/2 (pSer[217]/pSer[221], cat. no. 9192), phospho-ERK1/2 (pThr[202]/pTyr[204], cat. no. 9101), total ERK1/2 (cat. no. 9107), and phospho-Elk-1 (pSer[383], cat. no. 9181) were acquired from Cell Signaling Technology (Danvers, MA). U0126 and ERK inhibitor (CAS 1049738-54-6) were purchased from EMD Millipore/Calbiochem (Billerica, MA). PDGF-BB and FGF-2 were acquired from Peprotech (Rocky Hill, NJ). Sequencing grade trypsin was purchased from Promega (Madison, WI). Except where noted otherwise, all other reagents were obtained from Sigma Aldrich (St Louis, MO) or Fisher Scientific (Rockford, IL).

### Live-cell microscopy

Cells were detached with a brief trypsin-EDTA treatment and sus-pended in imaging buffer (20 mM HEPES pH 7.4, 125 mM NaCl, 5 mM KCl, 1.5 mM $MgCl_2$, 1.5 mM $CaCl_2$, 10 mM glucose and 2 mg/ml fatty-acid-free bovine serum albumin). After centrifugation at $100 \times g$ for 3 min, the cells were resuspended in imaging buffer and counted using a Beckman Coulter Counter. Adhesive surfaces were prepared on clean, sterile $25 \times 25$ mm glass cover slips (Fisher Scientific, Pittsburgh, PA), which were coated with poly-D-lysine (100 µg/ml) overnight at 4°C, washed with deionized, sterile water and dried. To make a chamber for the cells, a Teflon ring was attached to the cover slip, and 1 ml of cell suspension containing $3 \times 10^4$ cells (roughly 100 cells/mm²) was added. Cells were allowed to spread and were serum-starved in imaging buffer for 4 h prior to imaging. Mineral oil was layered on top of the buffer to

prevent evaporation during the experiment. Growth factors and inhibitors were diluted in the same buffer before adding to the cells.

Images were acquired using epifluorescence microscopy. A 50-W mercury arc lamp was used to excite EGFP, mCherry and Cerulean proteins with 480/20-nm, 572/23-nm and 436/20-nm excitation filters, respectively. Emission peaks of EGFP, mCherry, Cerulean and Venus fluorophores were gated using 515/30-nm, 630/60-nm, 480/40-nm and 540/30-nm bandpass filters, respectively (filter sets from Chroma Technology, Bellows Falls, VT). A 40 × water immersion objective (Zeiss Achroplan, 0.8 NA) and 0.63 × camera mount were used. Digital images were acquired using a Hamamatsu ORCA ER cooled charge-coupled device (CCD) (Hamamatsu, Bridgewater, NJ) and MetaMorph software (Universal Imaging, West Chester, PA). In each image, the intensity of an acellular region was defined as background and subtracted from the intensity of each pixel prior to further analysis. Binary masks of the nuclear and non-nuclear regions of each cell were applied to calculate mean intensity values of GFP-ERK1 or mCherry-ERK2 using custom codes in MATLAB (MathWorks, Natick, MA). In the case of mCherry-ERK2 co-expressed with one of the EKAR probes, the nuclear region was defined by the presence (nuclear EKAR) or absence (cytosolic EKAR) of Cerulean fluorescence. The average nuclear ERK intensity is normalized by the average ERK intensity of the whole cell for each frame, which accounts for a moderate degree of photobleaching; this quantity, measured as a function of time, is normalized again by its pre-stimulation value. For the EKAR biosensors, the ratio of average acceptor and donor (Venus/Cerulean) intensities was calculated as a relative measure of FRET signal for each image. Like the measurement of ERK nuclear translocation, EKAR FRET is normalized by its pre-stimulation value.

The time scale associated with the decay of nuclear ERK or EKAR following MEK inhibition was defined as $t_{1/2}$, the elapsed time after addition of the MEK inhibitor when the value of the measurement had decayed halfway between the value measured just prior to inhibition and that of the pre-stimulus baseline. For each cell, the time interval between successive frames during which this event occurred was identified, and the $t_{1/2}$ value was estimated by linear interpolation.

**Quantitative immunoblotting, immunofluorescence, and mass spectrometry**

Cells were serum-starved for 4 h prior to stimulation. Detergent lysates were prepared for quantitative immunoblotting, and immunoblots were performed using enhanced chemiluminescence, as described previously (Park *et al*, 2003). The BioRad Fluor S-Max system, which gives a linear response with respect to light output, was used, and band intensity was quantified using local background subtraction. The data were first normalized by an appropriate loading control and then further "trend" normalized to evaluate the consistency of relative kinetic trends across independent experiments, according to the procedures described in detail previously (Wang *et al*, 2009).

**Liquid chromatography-tandem mass spectrometry (LC/MS/MS) analysis**

Cell lysates were processed and trypsin-digested, and phosphopeptides were enriched by immobilized metal affinity chromatography

(IMAC), essentially as described previously (Chien *et al*, 2011). Briefly, 250 μg of each peptide digest was spiked with 5 μg of β-casein digest as an internal standard. Each sample was loaded onto 100 μl of Fe-NTA agarose slurry and washed with 200 μl of 2% acetic acid. A more stringent wash was performed with 200 μl of 74:25:1 100 mM NaCl/acetonitrile/acetic acid (v/v/v), followed by a wash with 100 μl of $H_2O$. Retained peptides were eluted with 200 μl of 5% $NH_4OH$, then immediately acidified to pH 3 with formic acid. The eluted peptides were dried using vacuum centrifugation and solubilized in 0.1% formic acid.

LC/MS/MS analyses were performed for biological triplicates using an Easy nLC 1000 liquid chromatograph coupled to an LTQ Orbitrap Elite mass spectrometer (Thermo Fisher Scientific, Bremen, Germany). Samples were injected onto a PepMap C18 5 μm trapping column (Thermo-Dionex, Sunnyvale, CA) then separated by in-line gradient elution onto a 75 μm id × 15 cm capillary in-house packed with 1.7 μm BEH C18 stationary phase (Waters Corp., Milford, MA). The linear gradient was carried out from 5 to 40% mobile phase B over 40 min at a 300 μl/min flow rate, where mobile phase A was 0.1% formic acid in 2% acetonitrile and mobile phase B was 0.1% formic acid in acetonitrile. The Orbitrap Elite was operated in data-dependent mode where the seven most intense precursors at 60k resolving power (at *m/z* 400) were selected for subsequent collision-induced dissociation (CID) fragmentation. The normalized collision energy was set to 35% for CID. For internal mass calibration, the ion at *m/z* 445.120025 was used as the lock mass. Monoisotopic precursor selection was enabled, and precursors with unassigned charge or a charge state of +1 were excluded. Fragmented precursor masses were excluded from further selection for 60 s.

The raw data files were processed using Proteome Discoverer version 1.3 (Thermo Fisher Scientific, San Jose, CA). Peak lists were searched against a forward and reverse *Mus musculus* UniProt database (UniprotKB release 2012_10) appended with bovine casein proteins using both Mascot (Matrix Science, UK) and Sequest (Thermo Fisher Scientific, San Jose, CA). The following parameters were selected to identify tryptic peptides for protein identification: 10 ppm precursor ion mass tolerance; 0.6 Da product ion mass tolerance; a maximum of two missed cleavages; carbamidomethylation of Cys was set as a fixed modification; oxidation of Met and phosphorylation of Ser, Thr, and Tyr were set as variable modifications. Searched results were filtered with the integrated Percolator node using an FDR < 5%. Phosphorylation site probabilities were calculated by the integrated phosphoRS node, and phosphorylation sites with a probability < 75% were discarded. Relative quantification based on precursor ion intensities of ERK2 phosphopeptides (+3 charge state only) was performed using Sieve v2.0 (Thermo Fisher Scientific, San Jose, CA) for the results shown in Fig 1; subsequent analysis shown in Supplementary Fig S2 was performed using Skyline v1.4 (Schilling *et al*, 2012). These data were trend-normalized in the same manner as the immunoblotting data.

**Immunofluorescence**

Immunofluorescence was carried out as follows. PDGF-stimulated cells were washed twice with phosphate-buffered saline (PBS) before fixation and permeabilization in CSK buffer (2.5% non-buffered formaldehyde and 0.5% Triton X-100 in 10 mM HEPES

(pH 7.0), 100 mM NaCl, 300 mM sucrose, 3 mM MgCl$_2$, 5 mM EGTA). After washing with PBS, the cells were stained with primary and Alexa 568-conjugated secondary antibodies, washed with PBS, and mounted on glass slides with Citifluor medium (Citifluor Ltd., London, UK). Images were acquired using a Zeiss LSM 710 laser scanning confocal microscope.

### Formulation and analysis of kinetic models

Kinetic models comprised of ordinary differential equations in time were constructed based on known or plausible signaling mechanisms, as described in detail in Supplementary Text S1. The models have the following common features, which collectively constitute a core model. MEK and ERK are each activated, in sequence, by two-site phosphorylation, and PDGF stimulation is modeled as a step increase in MEK phosphorylation rate. Activated ERK governs negative feedback through desensitization of MEK phosphorylation. These processes are assumed to occur in the cytosol, where MEK is strictly localized (Fukuda *et al*, 1996; Burack & Shaw, 2005), and their treatment in the model follows from previous work (Cirit *et al*, 2010) with judicious simplifications. To this we added nucleocytoplasmic shuttling of ERK, with translocation of diphosphorylated ERK into the nucleus. While in the nucleus, ERK is dephosphorylated by nuclear phosphatases, and it is exported back to the cytosol irrespective of its phosphorylation state (Horgan & Stork, 2003).

Building on this core model, our final model also includes interactions between diphosphorylated ERK and protein substrates in the cytosol and nucleus, which serve to anchor ERK in each compartment (Burack & Shaw, 2005; Lidke *et al*, 2010; Kim *et al*, 2011b). For simplicity we assume that one substrate (or a class of substrates with similar kinetic properties) dominates the interaction with ERK in each compartment, and the model accounts for phosphorylation of the substrate (by ERK) and its dephosphorylation. The similarities and differences between this model and that of Hirashima, who considered the combined influence of substrate interactions and nucleocytoplasmic shuttling at steady state (Hirashima, 2012), are enumerated in Supplementary Text S1.

A different model was formulated to test an alternative mechanism for kinetically decoupling nuclear translocation and nuclear activity of ERK, namely the release of ERK from the shuttling protein, Importin-7. The transition from Importin-bound to free, active ERK was modeled simply as a first-order process. This modification of the model was assessed instead of, and also in combination with, the substrate-binding model. All model variations are described in detail in Supplementary Text S1.

The approach used to fit models to kinetic data has been described in detail previously (Cirit *et al*, 2010; Cirit & Haugh, 2012) and is summarized in Supplementary Text S1. Briefly, it uses a Monte Carlo/simulated annealing-based algorithm to generate a large ($n = 10\,000$) ensemble of "good" parameter sets rather than one "best" fit. Starting with initial parameter values, the model output is computed and aligned with the data, and the quality of fit is evaluated. Next, new values are chosen according to normal distributions centered on the previous values; the width of the distribution is a parameter of the algorithm. If specified error criteria are met, the new parameter set is accepted and used as the nexus for choosing the next parameter values; otherwise, the procedure is repeated with the previous parameter set. If the starting parameter set

(determined based on initial trials) is nearly optimal, then the parameter sets generated in this manner constitute an ensemble that fit the data almost equally well. After compiling the ensemble, the model output is recalculated for each parameter set, and at each time point, an ensemble mean and standard deviation may be calculated.

**Supplementary information** for this article is available online: http://msb.embopress.org

### Acknowledgements

The authors wish to thank Stanislav Shvartsman (Princeton Univ.) for helpful discussions. This work was supported by the National Institutes of Health, grant no. R01GM088987.

### Author contributions

SA, KGG, LEE, MBG, and JMH designed research; SA, KGG, LEE, and AR performed research; SA, KGG, LEE, MBG, and JMH analyzed the results; MC provided valuable tools and expertise; SA and JMH wrote the paper with input from all other authors.

### Conflict of interest

The authors declare that they have no conflict of interest.

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
