## [Review Process File · Molecular Systems Biology]

Data-driven modeling reconciles kinetics of ERK phosphorylation, localization, and activity states

Shoeb Ahmed, Kyle G. Grant, Laura E. Edwards, Anisur Rahman, Murat Cirit, Michael B. Goshe and Jason M. Haugh

Corresponding author: Jason Haugh, North Carolina State University

Review timeline:

Submission date:	18 July 2013
Editorial Decision:	21 August 2013
Revision received:	12 November 2013
Editorial Decision:	10 December 2013
Revision received:	16 December 2013
Accepted:	18 December 2013

Transaction Report:

1st Editorial Decision

21 August 2013

Thank you again for submitting your work to Molecular Systems Biology. We have now heard back from the three referees who agreed to evaluate your manuscript. As you will see from the reports below, the reviewers acknowledge that your work is addressing a potentially interesting topic. Nevertheless, they raise a series of concerns, which should be convincingly addressed in a revision of the manuscript.

Several of the reviewers' comments refer to the need to provide additional controls and to better document/clarify several points throughout the manuscript. However, the reviewers also point out that additional experimentation and computational analyses are required in order to strengthen the main conclusions. Without repeating all the points listed below, among the more fundamental issues are the following:

- Including data from a larger number of cells is required to demonstrate a homogeneous cell behavior.
- Demonstrating the generality of the presented results to the activation of the ERK pathway by factors other than PDGF will significantly enhance the impact of the main findings.

On a more editorial level, we would like to encourage you to include the source data for the figures that show essential quantitative information. (Additional information is available in the "Guide for Authors" section in our website at <http://www.nature.com/msb/authors/index.html>)

If you feel you can satisfactorily deal with these points and those listed by the referees, you may wish to submit a revised version of your manuscript. Please attach a covering letter giving details of the way in which you have handled each of the points raised by the referees. A revised manuscript will be once again subject to review and you probably understand that we can give you no guarantee

at this stage that the eventual outcome will be favorable.

Reviewer #1:

The authors address the mechanism regulating ERK phosphorylation combining its dynamics and localization in relation to its activity. In this regard, the authors combine biochemical and live-cell imaging data with mathematical modeling. The authors show that growth factor induced ERK activation and nuclear translocation is transient with a high degree of attenuation. By employing an ERK activity fluorescent reporter construct, the authors show that the ERK activity is rather sustained, especially in the nucleus. By generating a kinetic mathematical model of ERK activation and cytosolic-nuclear cycling, the authors suggest that ERK-substrate interactions regulate the availability of free active ERK. Therefore, ERK-substrate interactions could buffer the ERK activity at high ERK phosphorylation degrees.

The work is of high interest as it demonstrates a deviation of ERK activity from ERK phosphorylation. Additional controls regarding both the biochemical results as well as the mathematical modeling would strengthen the findings.

Major comments:

- 1) The discrepancies of double-phosphorylated ERK and nuclear ERK accumulation on one hand and ERK activity as measured by the ERK FRET sensor on the other hand is very interesting, as double-phosphorylated ERK is generally regarded as synonymous with active ERK. It is of therefore of vital importance that this effect is validated with an alternative measurement. Unfortunately, the measurement of Elk1 phosphorylation is not as conclusive as stated by the authors, as the kinetics are in-between "active" and double-phosphorylated ERK (compare Fig. 2G, right panel with Figs. 2E, F). The measurement of other ERK substrate and/or in vitro kinase assays are required to clarify this point.
- 2) Different model structures have been compared to demonstrate that ERK-substrate interactions are required for the above-mentioned difference between ERK phosphorylation and activity. Somewhat unsurprising, the only models that can describe this effect harbor an ERK species in the cytoplasm and the nucleus that is double-phosphorylated, but inactive. In the other model (Fig. S5), the species double-phosphorylated ERK and active ERK are identical. Other mechanisms to obtain a double-phosphorylated but inactive ERK should also be explored.
- 3) The data on Elk1 phosphorylation should also be included for modeling. It would be interesting if the kinetics matches the p_n variable.
- 4) Figure 6: Why do the authors apply $S = 0.5$ for a weaker signal, instead of using 0.03 corresponding to the experimental data? Also, the authors should provide the reader with more insight about the "adjustment of the submaximal S value".

Minor comments:

- 1) In some figures, only data of either ERK1 or ERK2 is shown. The authors should display results for both ERK isoforms as they might affect the cellular behavior differently or comment on the redundancy of the two ERK isoforms.
- 2) Which ERK inhibitor was used?
- 3) The authors should clarify whether Figure 2G depicts the Elk-1 activity (as stated in the figure) or Elk-1 phosphorylation status (as indicated in the text).
- 4) Some statements are not sufficiently discussed, e.g. when the authors challenge the common perspective of rate-controlling process affecting ERK dynamics (page 8/9) or argue about the common assumption that nuclear translocation of ERK tracks its phosphorylation and enzymatic

activity. This holds true for the 'irreconcilable conflict' regarding the EKAR measurements (page 10). A more detailed explanation will facilitate a more comprehensive understanding of the argumentation.

5) Kinetic modeling: there is a model artifact accounting for the very fast peak (< 1 min) of the MEK kinetics (Fig. 5). The kinetic rate constants should be checked and the artifact should be removed.

6) Figure 2B: the labeling on the y axis has to be corrected: "Cytosol"

7) Figure 5: a better description of the figure is necessary. Possibly adding letters to the panels corresponding to specific substrate buffering strength values would be beneficial for the comprehension of the figure.

8) The authors should be consistent with their wording, e.g. 'ERK' or 'ERK1/2' and 'PDGF' or 'PDGF-BB'. Similarly, the authors should not alternate between e.g. 'biphosphorylated', 'dually-phosphorylated' and 'diphosphorylated' ERK.

Reviewer #2:

In their manuscript, Ahmed et al. present a new understanding of the kinetics of ERK activation and downstream activity. Making use of localized live-cell reporters of ERK activity, they are able to refine previous observations of the time line of ERK activation and inactivation following growth factor (here PDGF) treatment, and show that nuclear ERK activity lags the translocation of ERK to the nucleus. Based on these data, the authors develop an improved computational model of ERK activation in which their observations are explained by ERK's interaction (and phosphorylation of) nuclear substrate anchors.

The data is overall nicely presented and the paper is well written. However, there are some issues of concern, listed below, one in particular being that the conclusions of the paper are based on the observations of very few cells, and therefore should be strengthened by adding supporting evidence for the homogeneity of cellular behaviors or by adding data from additional cells. Finally one important question that is left unanswered is whether the results observed are specific to PDGF or whether they apply more globally to signaling by other growth factors activating the ERK pathway. Showing that the observed nuclear ERK/nuclear EKAR dynamics also occur with other growth factor stimulations could also greatly strengthen the impact of this manuscript.

Specific concerns and suggestions:

p.10 (bottom) - p.11 (top); One aspect that is left unexplained and therefore confusing at this point in the text is why nuclear ERK adapts, if, as stated in Text S1 "Cytosolic and nuclear ERK kinase activity reporter (EKAR) data are assumed to be proportional to c_{pp} and n_{pp}, respectively." Shouldn't we then expect nuclear ERK dynamics to be similar to nuclear EKAR dynamics and adapt slowly? Is this due to a different sensitivity threshold for ERK vs EKAR? Is this due to dephosphorylated ERK accounting for the peak of total nuclear ERK as it waits to exit the nucleus? This confusion is somewhat lifted after examining Figure 5, (where as I understood it, the green curves then represent what the EKAR activity would be fit to for each model). However it is still not quite satisfying as strictly speaking, the EKAR reporter really reports on phosphorylated substrate, not on free, dually phosphorylated ERK. One is left to wonder if a different explanation of the data could be arrived at if phospho-substrate had been considered in each of the models versions explored in the paper.

Regarding the EKAR reporters and the phenomena they report on:

A) Can the slow adaptation of the EKAR signal be explained by differences in sensitivity due to signal amplification (i.e. whereas low levels of ppERK or nuclear ERK might be undetectable, its phosphorylation products might still be abundant?)

B) Are the EKAR reporters (cytoplasmic and nuclear) shuttling? Could shuttling affect the interpretation of the data?

C) The okadaic acid experiment presented in Figure 3C should also be done for cytoplasmic EKAR; the earlier peak of EKAR signal in the cytoplasm could be due to the cytoplasmic reporter signal getting maxed out before the nuclear signal has reached its peak.

Figures 4, S5-S7. As the computational and experimental results are currently presented, it is not easy to appreciate the key qualitative successes and failures of each model implementation. The most interesting regions of the curves (parts F-H, 0-20 min) could be highlighted to make the comparisons easier. Also, what would be important to more strongly highlight that models in S5 and S6 *consistently* fail to match curves F-H (the mean +/- std curves are not visible on panels F, G and H - is this because they basically overlay the curve of the mean model behavior?). Perhaps a ratiometric descriptor (nuclear ERK activity/total nuclear ERK) would be a useful metric to present? Also the main text and the legend of Figure S7 could be construed as being contradictory ("... interactions did not demonstrably improve the quality of the fit..." vs. "...marginally improves the fit."

For the tested model predictions, it is implied, but not shown, that different models of ERK activation would predict qualitatively different outcomes. The paper would be strengthened by presenting additional computational results that show how the experimental tests of predictions based on the new model refute other model variants (Fig. 6 and 7).

Some statements of the discussion are not particularly well supported by data or computational results in this paper (and not accompanied by citations), e.g. "the analysis shows that measurable ERK dynamics are sensitive to relative reaction and translocation rates, which depend on input strength and also on the expression levels of various phosphatases and ERK substrates." Which figure presents a sensitivity analysis? Is this implied from Table S1? A boxplot showing validated parameter sets may be more appropriate.

Competitive interactions between ERK and its substrates were already shown to be important for regulating ERK phosphorylation, and presumably its activity, in *Drosophila* (cited as Kim, Y. 2011a and Kim, Y. 2011b); more emphasis should be put throughout the manuscript on what differentiates this study and what new conclusions it brings to the table.

The discussion could be improved by more broadly discussing how this new model of ERK activity regulation could impact the known biology of the ERK pathway. For example, does this have implications with regards to how cells respond to growth factors? Or implications for the therapeutic targeting of the pathway?

Other issues:

It would be important for the authors to clearly delineate how the imaging data is quantified for each reporter (e.g., is nuclear signal calculated from an integrated density, median density, average density?)

p.3. First paragraph of the introduction - "... (ERK) pathway is a dominant mode of controlling..."; with the term "dominant" having a very specific meaning in genetics, perhaps the term should be reconsidered here?

p.1 of Text S1; In the model presented, do MEK activation dynamics affect ERK activation dynamics, i.e. using a ramp function vs. a step function? The data presented, from a coarse time course, shows max activation at $t = 5$ min. Is there additional data or computational analysis to support using a step function at $t = 0$ min for the activation of MEK?

p. 1 of Text S1; Regarding the assumption that export from the nucleus does not depend on phosphorylation status; "... we confirmed that relaxing this particular assumption did not noticeably improve the quality of the fit." How is that assumption "relaxed"?

p. 4 of Text S1; Alignment factors are mentioned but not well defined. Which aspect of each time courses is adjusted by these alignment factors? These are all linear adjustments?

Table S1 would be easier to read if also reported as a boxplot. A boxplot would allow readers to quickly evaluate which parameters are better determined and which are more variable.

Fig. 1D and 2A, D, the figure should be annotated with a colorscale.

GFP-ERK1 is used (e.g. Figure 1D), but elsewhere ERK2 is emphasized, their equivalence should be supported by appropriate citations, especially to help readers with divergent research interests.

Figure 2G. The immunofluorescence data for phospho-Elk1 shows partial adaptation, not a lack of adaptation so in that regard seems to be qualitatively different from what is observed with nuclear EKAR.

Figure 6. The data presented is for a 33-fold jump in PDGF concentration, whereas the model uses only a 2-fold difference. Why does this discrepancy arise? Is there data supporting that 30 pM PDGF activated half as much MEK as 1 nM PDGF?

Figure 7. What are the distributions (mean, standard deviations) of the measured $t_{1/2}$ values? Is the observed difference between the two conditions statistically significant?

Figure 7A, B. A legend should be added for the different colored lines (or annotations completed in the figure legend).

Reviewer #3 :

This is a very nice paper that demonstrates the effect of substrate binding on ERK activation kinetics. It had previously been suggested that substrate binding by activated ERK could be responsible for an apparent attenuation or oscillation of its activity rather than the well-described negative feedback aspects of the pathway being responsible. This paper compellingly shows that substrate binding is likely to be a significant contributor to observed ERK kinetics. Furthermore, it provides a framework for understanding differences between the kinetics of ERK translocation, its effective activity and phosphorylation. The authors' model appears to nicely accommodate most of their "anomalous" results (or at least anomalous according to current dogma) and gratifyingly, a number of anomalous results that I have obtained in my own laboratory. Overall, I thought that this paper was very well written, very logical and provided significant biological insight.

I only had a few quibbles with their results and interpretation:

1. I had trouble with their assertion that negative feedback is not needed to obtain the transient nuclear translocation. During the parameter estimation for their model, they gave greatest weight to the nuclear translocation profile and the cytosolic and nuclear activity profile and lesser weight to the phosphorylation profiles. Although the fits appear to be better when substrate sequestration is included, all the models contained parameters for negative feedback. Testing models lacking negative feedback is necessary to make the statement that negative feedback is not needed.
2. They never adequately explain why there is a disparity between the kinetics of formation and loss of diphosphorylated ERK2 and mono-phosphorylated ERK2. They claim that it is predicted by their model, but I don't understand from their model why this should be so. They also only report on mono-phosphorylation on threonine in the main text. Why not include mono-phosphorylation on tyrosine?
3. It wasn't clear how background values were determined prior to subtraction or how binary masks of the nuclear and non-nuclear regions of each cell were created.
4. The statement at the bottom of page 7 that after 20 minutes "there was only a subtle dip in the phosphorylation level (Fig. 2G)" is an exaggeration. To my eye, it seems to be a substantial reduction. The extent of loss should be quantified and compared to results seen with the EKAR probes.
5. There is an inconsistency between the rates of loss of ERK activity following U0126 addition in figure 3B versus 7B. Why is this so? The much slower rate observed in Fig. 7B could affect their conclusion that activity ERK rapidly equilibrates between its substrates. This needs to be discussed and reconciled.
6. What was the ERK inhibitor used in Fig. S3?

Molecular Systems Biology manuscript MSB-13-4708

Data-driven modeling reconciles kinetics of ERK phosphorylation, localization, and activity states

by Ahmed et al.

Response to Reviewer comments

We sincerely thank all three Reviewers for their positive comments about our manuscript and for their suggestions of additional experiments and analyses to strengthen our story. We found the consistency of the comments between Reviewers refreshing (remarkable, actually). As outlined in detail below, we have endeavored to address all of their comments and have made corresponding changes to the manuscript. All changes in the revised manuscript are marked in blue.

We also wish to declare that we recently became aware of a theoretical analysis by Hirashima [*Math Biosci*, 239: 207-212 (2012)], who considered a model that includes both nucleocytoplasmic shuttling of ERK and ERK-substrate interactions. We assert that this work does not significantly diminish the novelty of ours. The central features of our paper are the experimental results and the manner in which the experiments are integrated with the modeling; Hirashima's paper is purely theoretical. Further, the primary emphasis of our analysis is on the temporal dynamics; Hirashima only considered the steady state, and therefore the goals and conclusions of the two papers do not overlap. We now cite Hirashima's paper in the Methods section of the main text, and in Text S1, we explain in detail the differences between Hirashima's models and ours.

Reviewer #1 (Remarks to the Author):

The authors address the mechanism regulating ERK phosphorylation combining its dynamics and localization in relation to its activity. In this regard, the authors combine biochemical and live-cell imaging data with mathematical modeling. The authors show that growth factor induced ERK activation and nuclear translocation is transient with a high degree of attenuation. By employing an ERK activity fluorescent reporter construct, the authors show that the ERK activity is rather sustained, especially in the nucleus. By generating a kinetic mathematical model of ERK activation and cytosolic-nuclear cycling, the authors suggest that ERK-substrate interactions regulate the availability of free active ERK. Therefore, ERK-substrate interactions could buffer the ERK activity at high ERK phosphorylation degrees.

The work is of high interest as it demonstrates a deviation of ERK activity from ERK phosphorylation. Additional controls regarding both the biochemical results as well as the mathematical modeling would strengthen the findings.

Major comments:

1) The discrepancies of double-phosphorylated ERK and nuclear ERK accumulation on

one hand and ERK activity as measured by the ERK FRET sensor on the other hand is very interesting, as double-phosphorylated ERK is generally regarded as synonymous with active ERK. It is of therefore of vital importance that this effect is validated with an alternative measurement. Unfortunately, the measurement of Elk1 phosphorylation is not as conclusive as stated by the authors, as the kinetics are in-between "active" and double-phosphorylated ERK (compare Fig. 2G, right panel with Figs. 2E, F). The measurement of other ERK substrate and/or in vitro kinase assays are required to clarify this point.

We see the Reviewer's point about the phospho-Elk-1 measurements. One subtle but important modification that we have made is to modify the Fig. 2 legend to make it clear that the scale of the time axis in Fig. 2G is longer than the others. Then, it should be clear that the early kinetics are much more similar to the nuclear EKAR response, as described under Results. We do agree, however, that the latter portion of the time course (≥ 30 minutes) is intermediate, and we suspect that it is this aspect that the Reviewer is referring to. This might be attributed to, e.g., changes in phosphatase activity acting on Elk-1. Qualitatively, we maintain that the apparent adaptation is modest (approximately 35%).

To strengthen this aspect of the story, we investigated the phosphorylation of another ERK substrate, as suggested by the Reviewer. Here we chose a strictly cytosolic substrate, MEK1, which is phosphorylated by ERK on a negative regulatory site, Thr292. As shown in new Supplementary Fig. S6, phosphorylation of MEK1 Thr292 responds rapidly to PDGF stimulation and shows little adaptation until after 60 minutes.

2) Different model structures have been compared to demonstrate that ERK-substrate interactions are required for the above-mentioned difference between ERK phosphorylation and activity. Somewhat unsurprising, the only models that can describe this effect harbor an ERK species in the cytoplasm and the nucleus that is double-phosphorylated, but inactive. In the other model (Fig. S5), the species double-phosphorylated ERK and active ERK are identical. Other mechanisms to obtain a double-phosphorylated but inactive ERK should also be explored.

We did test such a model as shown in current Supp. Fig. S8 (previous Fig. S6). In this model, nuclear translocation is a two-step process, with an intermediate ERK species that is counted as nuclear and phosphorylated but not in the active pool. This model was shown to fit the data poorly, with the explanation that this alternative buffering mechanism is not transient.

3) The data on Elk1 phosphorylation should also be included for modeling. It would be interesting if the kinetics matches the p_n variable.

A match between p_n and phospho-Elk-1 would implicate Elk-1 as a potential candidate for the substrate (or subset of substrates) that drives ERK dynamics; at the same time, it would be speculative to presume that any particular ERK

substrate such as Elk-1 is playing that role.

Hence, our approach was to compare the shape of the phospho-Elk-1 time course to those of the p_n variable and the free, active ERK (variable n_{pp}) predicted from the model. Hence, it was gratifying to see, as shown in the new Supplementary Fig. S10, that the phospho-Elk1 time course lies between the two extreme cases, as explained in the new paragraph appearing on p. 11 of the revised manuscript. We thank the Reviewer for suggesting this new analysis.

4) Figure 6: Why do the authors apply $S = 0.5$ for a weaker signal, instead of using 0.03 corresponding to the experimental data? Also, the authors should provide the reader with more insight about the "adjustment of the submaximal S value".

In previous publications from our laboratory, we have established that activation of the upstream signaling machinery feeding into MEK is saturated at submaximal receptor occupancy. As explained in Supplementary Text S1, the input S reflects these upstream processes in a coarse-grained manner; thus S is a saturable function of the PDGF concentration. As we now clarify in the text, $[PDGF] = 0.03$ nM has been established as approximately half-maximal (not simply submaximal) with respect to activation of the pathway, motivating the choice of $S = 0.5$ for that condition.

Regarding the latter point, we agree and have added the parenthetical comment, '... modest adjustment of the submaximal S value (e.g., in the range of $S = 0.3$ – 0.6 , because the saturable relationship between PDGF dose and S is approximate).'

Minor comments:

1) In some figures, only data of either ERK1 or ERK2 is shown. The authors should display results for both ERK isoforms as they might affect the cellular behavior differently or comment on the redundancy of the two ERK isoforms.

We agree with this point and now show the separate quantification of the ERK1 and ERK2 bands in our immunoblot data (addendum to Supp. Fig. S1), and we show further analysis of our mass spectrometry data quantifying phosphorylation of ERK1 alongside that of ERK2. These data show that the two isoforms show very similar phosphorylation kinetics, and we added mention to that effect in the revised manuscript. These results corroborate the previous indication that fluorescent protein-tagged ERK1 and ERK2 show similar nuclear translocation kinetics (Figs. 1 & 2).

2) Which ERK inhibitor was used?

Unfortunately, the EMD/Calbiochem catalog simply names the compound 'ERK inhibitor'. To remove this ambiguity, we have provided the CAS number in the

Materials & Methods section.

3) The authors should clarify whether Figure 2G depicts the Elk-1 activity (as stated in the figure) or Elk-1 phosphorylation status (as indicated in the text).

We thank the Reviewer for pointing out this error. Fig. 2 has been fixed to indicate phosphorylation rather than activity.

4) Some statements are not sufficiently discussed, e.g. when the authors challenge the common perspective of rate-controlling process affecting ERK dynamics (page 8/9) or argue about the common assumption that nuclear translocation of ERK tracks its phosphorylation and enzymatic activity. This holds true for the 'irreconcilable conflict' regarding the EKAR measurements (page 10). A more detailed explanation will facilitate a more comprehensive understanding of the argumentation.

We see the Reviewer's points here and have endeavored to clarify each of those concepts in the revised manuscript (regarding the 'common assumption', we clarified this point in the Introduction, where it is first broached).

5) Kinetic modeling: there is a model artifact accounting for the very fast peak (< 1min) of the MEK kinetics (Fig. 5). The kinetic rate constants should be checked and the artifact should be removed.

We see the Reviewer's point, although we would hesitate to call it a model artifact. The result in question concerns the freedom of the model fitting algorithm to generate output such that the MEK phosphorylation might peak much earlier and with a much higher magnitude than is reflected in the corresponding data. This is due to the limited temporal resolution of the MEK phosphorylation data. We argue that this is not a feature of the fit that drives the conclusions of the study. Although it is true that the fitting algorithm tends to favor a high MEK phosphorylation peak to 'squeeze the last ounce' out of the fit, there is a healthy degree of variability in this part of the fit that is less constrained by the available data.

New analysis performed in response comment #1 of Reviewer #3 provides a control of sorts for this fitting issue. We refit the ERK data using a model variation in which MEK activity is held constant. As shown in the new Supplementary Fig. S11, this variation fits key aspects of the data just as well if not better than the regular model (phospho-ERK immunoblot, ERK nuclear translocation, nuclear EKAR), while fitting other aspects not as well (cytosolic EKAR most notably).

6) Figure 2B: the labeling on the y axis has to be corrected: "Cytosol"

The labeling in this case is correct. As explained in the legend and corresponding section of the Results, each of the cytosolic and nuclear EKAR

FRET probes were multiplexed with mCherry-tagged ERK2; in each case, the nuclear translocation of mCherry-ERK2 was monitored. So, in Fig. 2B, 'Nuclear' is correct.

7) Figure 5: a better description of the figure is necessary. Possibly adding letters to the panels corresponding to specific substrate buffering strength values would be beneficial for the comprehension of the figure.

We recognize that Fig. 5 is complex, as it consolidates our understanding of how the model fits the data set. The Reviewer's suggestion about adding letters is well taken, but in our opinion, that scheme would not convey the information any more readily. Instead we have modified both the Fig. 5 legend and the Results text to make it clear that the relationships between plots are indicated by arrows.

8) The authors should be consistent with their wording, e.g. 'ERK' or 'ERK1/2' and 'PDGF' or 'PDGF-BB'. Similarly, the authors should not alternate between e.g. 'biphosphorylated', 'dually-phosphorylated' and 'diphosphorylated" ERK.

We have fixed all of these instances in the revised manuscript text, except where we felt that a distinction was warranted in the Materials & Methods or figure captions.

Reviewer #2 (Remarks to the Author):

In their manuscript, Ahmed et al. present a new understanding of the kinetics of ERK activation and downstream activity. Making use of localized live-cell reporters of ERK activity, they are able to refine previous observations of the time line of ERK activation and inactivation following growth factor (here PDGF) treatment, and show that nuclear ERK activity lags the translocation of ERK to the nucleus. Based on these data, the authors develop an improved computational model of ERK activation in which their observations are explained by ERK's interaction (and phosphorylation of) nuclear substrate anchors.

The data is overall nicely presented and the paper is well written. However, there are some issues of concern, listed below, one in particular being that the conclusions of the paper are based on the observations of very few cells, and therefore should be strengthened by adding supporting evidence for the homogeneity of cellular behaviors or by adding data from additional cells.

We appreciate the Reviewer's concern but argue that the issue of cell-to-cell heterogeneity was adequately addressed by showing kinetic traces of the individual cells for two of the key experiments (current Supplementary Figures S4 and S12).

As for the day-to-day reproducibility of the results, we note that the key live-cell responses to PDGF were each repeated in separate sets of experiments, with

consistent outcomes. Nuclear translocation of ERK2 was repeated in Fig. 2B&E, Fig. 7A, and Supp. Fig. S12B (previous Fig. S8B), and similar results were seen for ERK1 in Fig. 1D. Cytosolic EKAR was repeated in Fig. 2C, Fig. 3A, and Fig. 3C (the latter is new data provided in response to Reviewer #1), and nuclear EKAR was repeated in Fig. 2F, Fig. 3B&D, and Fig. 7B.

Finally one important question that is left unanswered is whether the results observed are specific to PDGF or whether they apply more globally to signaling by other growth factors activating the ERK pathway. Showing that the observed nuclear ERK/nuclear EKAR dynamics also occur with other growth factor stimulations could also greatly strengthen the impact of this manuscript.

We agree with the Reviewer's suggestion and have repeated the nuclear ERK and nuclear EKAR experiments under FGF-2 stimulation, showing qualitatively similar results. These data are shown in new Supp. Fig. S5.

Specific concerns and suggestions:

p.10 (bottom) - p.11 (top); One aspect that is left unexplained and therefore confusing at this point in the text is why nuclear ERK adapts, if, as stated in Text S1 "Cytosolic and nuclear ERK kinase activity reporter (EKAR) data are assumed to be proportional to c_{pp} and n_{pp} , respectively." Shouldn't we then expect nuclear ERK dynamics to be similar to nuclear EKAR dynamics and adapt slowly? Is this due to a different sensitivity threshold for ERK vs EKAR? Is this due to dephosphorylated ERK accounting for the peak of total nuclear ERK as it waits to exit the nucleus? This confusion is somewhat lifted after examining Figure 5, (where as I understood it, the green curves then represent what the EKAR activity would be fit to for each model). However it is still not quite satisfying as strictly speaking, the EKAR reporter really reports on phosphorylated substrate, not on free, dually phosphorylated ERK. One is left to wonder if a different explanation of the data could be arrived at if phospho-substrate had been considered in each of the models versions explored in the paper.

To alleviate the confusion, we have added a key sentence on p. 11 of the revised manuscript, 'Thus, the adaptation of total ERK in the nucleus reflects a reduction in the fraction of diphosphorylated ERK that is substrate-bound (and anchored in the nucleus) versus freely available.' This explains why total nuclear ERK is different from nuclear EKAR; the former includes active ERK that is free to interact with EKAR but also ERK that is in other states, most of which is in complex with the high-affinity substrate(s). Put in terms of the model, as a function of time there is a shift between the ERK-substrate complex (n_p) and the free, active ERK (n_{pp}) (shown in Fig. 5 as noted by the Reviewer). The slow export of dephosphorylated ERK is not a viable explanation that fits the data; in that case we would not expect the nuclear EKAR to peak later than total nuclear ERK.

The paragraph that follows the statement mentioned above, which was added to address a comment raised by Reviewer #1, provides additional explanation of the

conceptual model leading up to the description of Fig. 5.

The final part of the comment concerns the nature of the EKAR biosensor, which reports its own phosphorylation by ERK. We assume that this readout is faithful to the free, active ERK because of the rapid kinetics of its dephosphorylation, apparent from Fig. 3. Thus, the alternative explanation that nuclear EKAR kinetics are delayed relative to nuclear ERK, which we worried about initially, was ruled out by experiment. This is why we devoted a whole subsection of the Results to this point before diving into the modeling.

Regarding the EKAR reporters and the phenomena they report on:

A) Can the slow adaptation of the EKAR signal be explained by differences in sensitivity due to signal amplification (i.e. whereas low levels of ppERK or nuclear ERK might be undetectable, its phosphorylation products might still be abundant?)

If we understand the comment correctly, there are two parts of the question to address. We are suggesting that the answer to the latter part is yes; the ‘low’ level of ppERK at steady state is driving significant levels of phosphorylated substrates. We think that the new Fig. S10 (provided in response to Reviewer #1) helps to explain that concept. The other side of that coin is that the ‘high’ level of ppERK reflects the condition at early times, when little of the substrate has been phosphorylated, and there is intense competition for active ERK that is ordered according to the buffering strengths of the various substrates.

Regarding the first part of the question, the answer is no. If it were a matter of sensitivity, that argument clearly breaks down for total nuclear ERK versus nuclear EKAR. If nuclear EKAR were insensitive to a decrease in total ERK in the nucleus, the EKAR signal would be expected to plateau before the peak in total ERK is achieved, not after.

B) Are the EKAR reporters (cytoplasmic and nuclear) shuttling? Could shuttling affect the interpretation of the data?

The cytosolic and nuclear EKAR probes contain nuclear exclusion and nuclear localization sequences, respectively. Although we acknowledge that shuttling must occur (e.g., nuclear EKAR can passively exit the nucleus, but it rapidly returns), Fig. 2 shows strict localization for each of the biosensors, confirming that the fraction of each biosensor in the unintended compartment is undetectable. One can also argue that the clear temporal separation between the cytosolic and nuclear EKAR responses shows that ‘mixing’ of the signals due to shuttling is not an issue.

C) The okadaic acid experiment presented in Figure 3C should also be done for cytoplasmic EKAR; the earlier peak of EKAR signal in the cytoplasm could be due to the cytoplasmic reporter signal getting maxed out before the nuclear signal has reached its peak.

We agree and have acquired additional data that rule out this possibility (new panel in revised Fig. 3). Like the previously shown results with nuclear EKAR, cytosolic EKAR shows sequentially increased signal in response to PDGF stimulation followed by okadaic acid treatment.

Figures 4, S5-S7. As the computational and experimental results are currently presented, it is not easy to appreciate the key qualitative successes and failures of each model implementation. The most interesting regions of the curves (parts F-H, 0-20 min) could be highlighted to make the comparisons easier.

We agree and have added text to both captions of the supplementary figures that present the failed models (Figs. S7&S8 of the revised manuscript, which were Figs. S5&S6 in the original manuscript).

Also, what would be important to more strongly highlight that models in S5 and S6 *consistently* fail to match curves F-H (the mean +/- std curves are not visible on panels F, G and H - is this because they basically overlay the curve of the mean model behavior?). Perhaps a ratiometric descriptor (nuclear ERK activity/total nuclear ERK) would be a useful metric to present?

We agree with the comment about the consistency of those models' failures (the mean +/- std curves are indeed close together), and indeed we have incorporated this language in the aforementioned captions.

We appreciate the suggestion of presenting the ratio of nuclear ERK activity/total nuclear ERK. From the computed output of the failed models, this ratio is constrained to be more or less flat for most of the time course. In other words, the failed model fits are constrained such that the ERK activity and total nuclear ERK time courses have approximately the same shape, as we now note in the Fig. S7 caption. In our opinion, this makes the point, whereas showing the ratio as a function of time would complicate the presentation (for one thing, the ratio diverges at $t = 0$).

Also the main text and the legend of Figure S7 could be construed as being contradictory ("... interactions did not demonstrably improve the quality of the fit..." vs. "...marginally improves the fit."

We agree and have fixed the legend (current Fig. S9) accordingly.

For the tested model predictions, it is implied, but not shown, that different models of ERK activation would predict qualitatively different outcomes. The paper would be strengthened by presenting additional computational results that show how the experimental tests of predictions based on the new model refute other model variants (Fig. 6 and 7).

We agree in principle with this point; however, we do not currently have another model that fits the ‘training’ data set well, nor a plausible suggestion from the Reviewer of another mechanism that might. The alternate model variations that we did test are already refuted by virtue of their poor fits (for reasons that we have clarified in response to comments above); we do not see a rationale for using those models to make predictions.

We did consider alternate model schemes, or combinations thereof. For example, we thought of scenarios where explicit accounting of ERK interactions with phosphatase(s) might be sufficient to explain the data. At one extreme, the diphosphorylated ERK is saturated by the phosphatase, preventing the interaction of ERK with the EKAR sensor and other substrates; however, such saturation would only become more, not less, pronounced as the level of diphosphorylated ERK adapts with time. It also cannot predict the Fig. 7 result. At the other extreme, the phosphatase is saturated by di- and mono-phosphorylated ERK. At least qualitatively, this scenario can explain the Fig. 7 result but nothing else. The same is true of the possibility that ERK phosphatase activity is upregulated with time following stimulation (as documented for MKP-1, for example). In our opinion, modeling these ‘dead on arrival’ scenarios would complicate the presentation of the manuscript.

In the revised manuscript, we do now mention one variation of the current scheme (in both the Results and Discussion), namely the possibility that phosphorylation of the high-affinity substrate by ERK promotes degradation of that substrate. In that scenario, the transient buffering concept is the same; the difference is that the substrate would not need to be phosphorylated to a high stoichiometry.

Some statements of the discussion are not particularly well supported by data or computational results in this paper (and not accompanied by citations), e.g. "the analysis shows that measurable ERK dynamics are sensitive to relative reaction and translocation rates, which depend on input strength and also on the expression levels of various phosphatases and ERK substrates." Which figure presents a sensitivity analysis? Is this implied from Table S1? A boxplot showing validated parameter sets may be more appropriate.

The Reviewer’s point is well taken, as it shows that the meaning of that sentence was not clearly conveyed. The interpretation espoused in the sentence in question is that the shape of the nuclear translocation time course reflects the collective rate of ERK activation and import into the nucleus versus the time scale of substrate phosphorylation in the nucleus (i.e., if the former is much faster than the latter, one will see a prominent peak; otherwise, not).

In response we have changed the sentence in question to read, ‘More generally, the analysis shows that measurable ERK dynamics are sensitive to the degree of buffering by ERK substrates, which reflects the expression level(s) of the highest

affinity substrate(s) (Fig. 5). The kinetics and extent of substrate phosphorylation, which depend on the expression of phosphatases that reverse the action of ERK, are just as important (Fig. S10).⁷ In our opinion this better frames the point that ERK dynamics should be expected to vary in a context-dependent manner.

Competitive interactions between ERK and its substrates were already shown to be important for regulating ERK phosphorylation, and presumably its activity, in *Drosophila* (cited as Kim, Y. 2011a and Kim, Y. 2011b); more emphasis should be put throughout the manuscript on what differentiates this study and what new conclusions it brings to the table.

We appreciate the Reviewer's point. Our citations to the work from Kim and colleagues reflect our high regard for the work and the manner in which it guided our thinking. As the Reviewer states, the work was performed using *Drosophila* as a model system, and hence its ultimate impact depends on the extent to which it generalizes to mammalian cells/organisms.

We have added two passages in the revised manuscript, one near the end of the Introduction and the other at the end of the first paragraph of the Discussion. In both instances we mention the advance of our work in the context of mammalian cells but do not draw too much attention to that aspect. What is more important in our view, which we hope is now reinforced, is that our work uniquely implicates how substrate interactions explain newly revealed temporal dynamics.

The discussion could be improved by more broadly discussing how this new model of ERK activity regulation could impact the known biology of the ERK pathway. For example, does this have implications with regards to how cells respond to growth factors? Or implications for the therapeutic targeting of the pathway?

We appreciate the Reviewer's point and have largely rewritten/reorganized the last paragraph of the Discussion. In the original manuscript, we mentioned the basic ideas of what the Reviewer is asking for, but in the revised version we have endeavored to present these points in a more deliberate manner.

Other issues:

It would be important for the authors to clearly delineate how the imaging data is quantified for each reporter (e.g., is nuclear signal calculated from an integrated density, median density, average density?)

We have now clarified in the methods section that the signal is calculated from an average intensity, and we have added other details regarding how the nuclear region was defined and how those data were normalized.

p.3. First paragraph of the introduction - "... (ERK) pathway is a dominant mode of

controlling..."; with the term "dominant" having a very specific meaning in genetics, perhaps the term should be reconsidered here?

We have changed 'dominant' to 'principal.'

p.1 of Text S1; In the model presented, do MEK activation dynamics affect ERK activation dynamics, i.e. using a ramp function vs. a step function? The data presented, from a coarse time course, shows max activation at t = 5 min. Is there additional data or computational analysis to support using a step function at t = 0 min for the activation of MEK?

As explained in more detail now in the revised Text S1, we have worked out those details in previous publications, showing that the inputs to MEK are saturated under maximal PDGF stimulation conditions (Wang et al., 2009, Cirit et al., 2010). Although this description is not perfect by any means, the goal was to make it as simple as possible while also fitting the MEK phosphorylation time course well enough. We certainly debated about what to do for the predictions related to the submaximal dose of PDGF, where the pathway is obviously less saturated. Ultimately we decided to err on the side of a simpler model and rely on the qualitative nature of the predictions rather than dusting off the far more complicated models of the upstream signaling in hopes of eking out a better quantitative match to the data.

p. 1 of Text S1; Regarding the assumption that export from the nucleus does not depend on phosphorylation status; "... we confirmed that relaxing this particular assumption did not noticeably improve the quality of the fit." How is that assumption "relaxed"?

To clarify we have reworded as follows. 'We confirmed that allowing different rate constants for export of the various phospho-forms did not noticeably improve the quality of fit.'

p. 4 of Text S1; Alignment factors are mentioned but not well defined. Which aspect of each time courses is adjusted by these alignment factors? These are all linear adjustments?

We have clarified this step of the algorithm in Text S1. They are indeed linear adjustments to rescale the model outputs to the arbitrary units of the data.

Table S1 would be easier to read if also reported as a boxplot. A boxplot would allow readers to quickly evaluate which parameters are better determined and which are more variable.

We appreciate the suggestion and have prepared such a boxplot, with two parts dividing the parameters into two, almost equal-sized groups according to the size of the box (i.e., according to the ratio of their third and first quartiles).

Fig. 1D and 2A, D, the figure should be annotated with a color scale.

We argue that a color scale would only be necessary had we not also shown the quantification for all cases.

GFP-ERK1 is used (e.g. Figure 1D), but elsewhere ERK2 is emphasized, their equivalence should be supported by appropriate citations, especially to help readers with divergent research interests.

This point was addressed in our response to Reviewer #1, minor comment #1. To summarize, we have shown by further analysis of our immunoblotting and mass spectrometry data that the kinetics of ERK1 and ERK2 are comparable in all respects.

Figure 2G. The immunofluorescence data for phospho-Elk1 shows partial adaptation, not a lack of adaptation so in that regard seems to be qualitatively different from what is observed with nuclear EKAR.

This point was addressed in our response to Reviewer #1, major comment #1. Our response to his/her major comment #3 is also relevant.

Figure 6. The data presented is for a 33-fold jump in PDGF concentration, whereas the model uses only a 2-fold difference. Why does this discrepancy arise? Is there data supporting that 30 pM PDGF activated half as much MEK as 1 nM PDGF?

This point was addressed in our response to Reviewer #1, minor comment #4.

Figure 7. What are the distributions (mean, standard deviations) of the measured $t_{1/2}$ values? Is the observed difference between the two conditions statistically significant?

We have added this information in the results text. The observed differences for both nuclear translocation and nuclear EKAR do indeed have very low p values by Student's t-test. We also added a description of how we estimated $t_{1/2}$ in the methods section.

Figure 7A, B. A legend should be added for the different colored lines (or annotations completed in the figure legend).

We have revised the figure to mark the times at which inhibition was imposed in the model.

Reviewer #3 (Remarks to the Author):

This is a very nice paper that demonstrates the effect of substrate binding on ERK activation kinetics. It had previously been suggested that substrate binding by activated

ERK could be responsible for an apparent attenuation or oscillation of its activity rather than the well-described negative feedback aspects of the pathway being responsible. This paper compellingly shows that substrate binding is likely to be a significant contributor to observed ERK kinetics. Furthermore, it provides a framework for understanding differences between the kinetics of ERK translocation, its effective activity and phosphorylation. The authors' model appears to nicely accommodate most of their "anomalous" results (or at least anomalous according to current dogma) and gratifyingly, a number of anomalous results that I have obtained in my own laboratory. Overall, I thought that this paper was very well written, very logical and provided significant biological insight.

I only had a few quibbles with their results and interpretation:

1. I had trouble with their assertion that negative feedback is not needed to obtain the transient nuclear translocation. During the parameter estimation for their model, they gave greatest weight to the nuclear translocation profile and the cytosolic and nuclear activity profile and lesser weight to the phosphorylation profiles. Although the fits appear to be better when substrate sequestration is included, all the models contained parameters for negative feedback. Testing models lacking negative feedback is necessary to make the statement that negative feedback is not needed.

The Reviewer makes an excellent point that one of the important implications of the paper might be taken as an unsupported claim. To remove this weakness, we performed an additional fit to the ERK data in which MEK activity was held constant (step change at time zero). As described now in the main text and shown in new Supplementary Fig. S11, this variation of the model successfully captures the adaptation of ERK phosphorylation and nuclear translocation without negative feedback.

We wish to stress that this should not imply that negative feedback is not important. As stated in the Discussion, it is clearly important for setting the quasi-steady activity of MEK that is ultimately achieved, and other groups have shown that it balances ultrasensitivity to assure a more linear input-output relationship.

2. They never adequately explain why there is a disparity between the kinetics of formation and loss of diphosphorylated ERK2 and mono-phosphorylated ERK2. They claim that it is predicted by their model, but I don't understand from their model why this should be so.

We see the Reviewer's point and have added the following explanation in our explanation of the fit shown in Fig. 4. 'A key aspect of the fit is that the total amount of diphosphorylated ERK (as measured in a cell lysate) includes both the free and substrate-bound pools. The fit shows that the kinetics of free, diphosphorylated ERK are actually better reflected in the accumulation of mono-phosphorylated ERK, for which the slow step is the liberation of substrate-bound

ERK followed by rapid dephosphorylation and, in the case of nuclear ERK, export to the cytosol.'

They also only report on mono-phosphorylation on threonine in the main text. Why not include mono-phosphorylation on tyrosine?

Quantification of both ERK2 phospho-forms (pTEY and TEpY) were shown in Supplementary Fig. S2, to which we have since added comparable data for the other ERK isoform, ERK1. These data are described in the main text, and for comparison we have also added the data for the TEpY form of ERK2 in Fig. 4D.

3. It wasn't clear how background values were determined prior to subtraction or how binary masks of the nuclear and non-nuclear regions of each cell were created.

We have added more explicit description of these methods (also in response to a similar comment by Reviewer #2).

4. The statement at the bottom of page 7 that after 20 minutes "there was only a subtle dip in the phosphorylation level (Fig. 2G)" is an exaggeration. To my eye, it seems to be a substantial reduction. The extent of loss should be quantified and compared to results seen with the EKAR probes.

We see the Reviewer's point, which echoes comments made by the other two Reviewers as well. To summarize our response, 1) we acknowledge the dip in the revised manuscript (incidentally, the value at 120 minutes is approximately 65% of the peak value, with the initial value subtracted from both) and note that this could be for reasons other than the dynamics of ERK; 2) new data show sustained phosphorylation of a cytosolic ERK substrate to complement the nuclear phospho-Elk-1; and 3) we now directly compare the phospho-Elk-1 kinetics to the modeling results and discuss the comparison.

5. There is an inconsistency between the rates of loss of ERK activity following U0126 addition in figure 3B versus 7B. Why is this so? The much slower rate observed in Fig. 7B could affect their conclusion that activity ERK rapidly equilibrates between its substrates. This needs to be discussed and reconciled.

We disagree with the Reviewer's observation. When U0126 was added after the response to PDGF had plateaued (46 minutes), the decay of nuclear EKAR showed $t_{1/2}$ values of ~ 2 minutes (note that the time resolution of the data is different: 2 min in Fig. 3 and 1 min in Fig. 7). The difference is modest in our view, considering that the experiments in common between the two figures were performed months apart. The clear difference was seen when the inhibitor was added earlier, prior to the establishment of a quasi-steady state. The slower decay under those conditions is consistent with the model, and thus constitutes a key prediction, as explained in the text.

6. What was the ERK inhibitor used in Fig. S3?

This ambiguity was pointed out by Reviewer #1 as well. Unfortunately, the EMD/Calbiochem catalog simply names the compound 'ERK inhibitor'. To remove this ambiguity, we have provided the CAS number in the Materials & Methods section.

2nd Editorial Decision

10 December 2013

Thank you again for submitting your work to Molecular Systems Biology. We have now heard back from the referee who accepted to evaluate your revised manuscript. As you will see, the referee thinks that the main concerns have been satisfactorily addressed and supports publication of the work, but raises a few minor points, which we would like to ask you to address in a revision of the manuscript.

Reviewer #2:

I am happy to see that in their revised manuscript, Ahmed and colleagues have addressed most of the concerns I had previously listed. The revised manuscript is strengthened by new data on both forms of ERK, on the impact of okadaic acid on cytosolic ERK activity, on MEK-Thr292 phosphorylation and on the response to FGF-2 stimulation. Furthermore, the new text does clarify points that were difficult to understand in the previous version and the new discussion more clearly delineates the new contributions and impact of the authors' work on the field.

Minor points:

p. 13 on the "maximal (1 nM) and roughly half-maximal (30 pM) doses of PDGF"; perhaps what would be even more helpful is if the authors defined what is maximized (is it the activation of MEK?) at 1 nM.

p. 15-16 on the discussion of the degree of buffering by ERK substrates. Here the authors note that the abundance (and affinity for ERK) of the ERK substrates will affect the ERK activity dynamics. One then might wonder whether the expression of EKAR - a new substrate of ERK, can significantly affect the dynamics observed and in itself cause the buffering effect that they observed. This might be worth noting in the discussion along with data in favor of the authors' interpretation (they also observed weak adaptation in two endogenous substrate by immunoblot in the parental cell line) and an estimate of the abundance of EKAR relative to other known substrates or ERK.

2nd Revision - authors' response

16 December 2013

December 16, 2013

Maria Polychronidou
Editor
Molecular Systems Biology

RE: Submission of re-revised manuscript MSB-13-4708RR

Dear Dr. Polychronidou,

On behalf of my co-authors I thank you for the reviews of our revised manuscript, "Data-driven modeling reconciles kinetics of ERK phosphorylation, localization, and activity states." We sincerely appreciate the Reviewers' attention to the revisions and the positive comments and constructive suggestions by Reviewer #2 that you transmitted to us. We agree with both of the points raised in the review and have addressed them in the re-revised manuscript, which we submit herewith for your consideration.

As we read the remaining points to be rather minor at this stage, we provide the explanation of the changes as an appendix to this covering letter below. Also, we chose not to mark the changes in the new version of the manuscript (to simplify submission of a clean Word file); the added passages are quoted verbatim below.

In addition to those changes, we have:

- moved the Supplementary Figure legends to the Supplementary Information
- Included a Conflict of Interest statement in the main article file
- provided a synopsis file with the requisite standfirst text and bullet points
- provided a thumbnail image for the table of contents.

We know that only high-quality articles are chosen for publication in Molecular Systems Biology. That said, we feel that this is an important paper in cell signaling and ask you to please consider its suitability as a featured article.

We declare that there are no competing commercial interests and that no part of this work is under consideration for publication elsewhere.

Sincerely,
Jason M. Haugh

Response to minor points by Reviewer #2

p. 13 on the "maximal (1 nM) and roughly half-maximal (30 pM) doses of PDGF"; perhaps what would be even more helpful is if the authors defined what is maximized (is it the activation of MEK?) at 1 nM.

Response:

We agree with the Reviewer's point and have revised the sentence in question to read, "First, we compared PDGF doses that yield maximal (1 nM) and roughly half-maximal (30 pM) phosphorylation of ERK (Supplementary Fig. S1)."

p. 15-16 on the discussion of the degree of buffering by ERK substrates. Here the authors note that the abundance (and affinity for ERK) of the ERK substrates will affect the ERK activity dynamics. One then might wonder whether the expression of EKAR - a new substrate of ERK, can significantly affect the dynamics observed and in itself cause the buffering effect that they observed. This might be worth noting in the discussion along with data in favor of the authors' interpretation (they also observed weak adaptation in two endogenous substrate by immunoblot in the parental cell line) and an estimate of the abundance of EKAR relative to other known substrates or ERK.

Response:

We understand the Reviewer's point and have revised and amended the passage in the Discussion to read, "More generally, the analysis shows that measurable ERK dynamics are sensitive to the degree of buffering by ERK substrates (Fig. 5). We note that expression of either of the EKAR substrates does not noticeably perturb ERK in our cells, as judged from the consistent nuclear translocation kinetics measured in parallel with EKAR responses or in the absence of EKAR expression. This observation underscores the conclusion that only the substrates with the highest buffering strengths affect ERK dynamics."